organometallic chemistry/mesoscopics/
environmental chemistry

antibiotic pollutant, MIL-53 (Fe), hollow
mesoporous carbon, ibuprofen adsorption

**Author for correspondence:**
Trinh Duy Nguyen
e-mail: ndtrinh@ntt.edu.vn

This article has been edited by the Royal Society of Chemistry, including the commissioning, peer review process and editorial aspects up to the point of acceptance.

# A hollow mesoporous carbon from metal-organic framework for robust adsorbability of ibuprofen drug in water

Thuan Van Tran[1], Duyen Thi Cam Nguyen[1,2],
Hanh T. N. Le[3], Oanh T. K. Nguyen[1], Vinh Huu Nguyen[1],
Thuong Thi Nguyen[1], Long Giang Bach[1]
and Trinh Duy Nguyen[1]

[1]NTT Hi-Tech Institute, and [2]Department of Pharmacy, Nguyen Tat Thanh University,
298–300A Nguyen Tat Thanh, Ward 13, District 4, Ho Chi Minh City 700000, Vietnam
[3]Institute of Hygiene and Public Health, 159 Hung Phu, Ward 8, District 8,
Ho Chi Minh City 700000, Vietnam

TVT, 0000-0001-6354-0379; DTCN, 0000-0002-3311-6945;
HTNL, 0000-0003-2354-269X; OTKN, 0000-0002-3137-4190;
VHN, 0000-0002-8756-396X; TTN, 0000-0003-0474-4317;
LGB, 0000-0003-1160-6705; TDN, 0000-0002-7241-1488

Herein, we described a tunable method for synthesis of novel hollow mesoporous carbon (MPC) via direct pyrolysis (800°C) of MIL-53 (Fe) as a self-sacrificed template. The structural characterization revealed a hollow, amorphous, defective and mesoporous MPC along with high surface area (approx. $200 \, m^2 \, g^{-1}$). For the experiments of ibuprofen adsorption onto MPC, effects of contact time, MPC dosage, ionic strength, concentration and temperature were systematically investigated. The optimal conditions consisted of pH = 3, concentration $10 \, mg \, l^{-1}$ and dose of $0.1 \, g \, l^{-1}$ for the highest ibuprofen removal efficiency up to 88.3% after 4 h. Moreover, adsorption behaviour, whereby chemisorption and monolayer controlled the uptake of ibuprofen over MPC, were assumed. Adsorption mechanisms including H-bonding, $\pi-\pi$ interaction, metal–oxygen, electrostatic attraction were rigorously proposed. In comparison to several studies, the MPC nanocomposite in this work obtained the outstanding maximum adsorption capacity ($206.5 \, mg \, g^{-1}$) and good reusability (5 cycles); thus, it can be used as a feasible alternative for decontamination of ibuprofen anti-inflammatory drug from water.

# 1. Introduction

Pharmaceutically bioactive compounds (PBCs) are widely consumed all over the world because of their crucial role played in protecting human's health from the attack of bacteria species, as well as exhibiting a wide range of biological activities (e.g. antifungal, anti-cancer, anti-tumour, anti-inflammatory, antioxidant, etc.) [1]. However, accumulation of these emerging micro-pollutants in treated wastewater is increasingly detected, resulting in adverse effects on some enzymatic, hormonal and genetic systems, and posing risks for the environment [2,3].

Ibuprofen, an emerging representative of non-steroidal anti-inflammatory drug, is one of the most widespread pharmaceuticals presenting in groundwater [4]. Chemically, this drug molecule, whose properties and structure are summarized in electronic supplementary material, table S1 and figure S1, is constructed from aromatic ring substituted with carboxylic acid (pK$_a$ value of 5) including 3 H-bonds (1 H-acceptor and 2 H-donors). In microorganisms, ibuprofen is rapidly metabolized in the form of hydroxyl- and carboxyl-ibuprofen [5]. Naturally, ibuprofen residues can derive from wastewater in the pharmaceutical industries, and partial excrement of medically treated humans and animals [6].

For concentration of IBU in wastewater, Miège et al. [7] reported a database to quantitatively assess the occurrence and removal efficiency of pharmaceuticals and personal care products in wastewater treatment processes from many scientific publications, in which IBU concentrations in the effluents leaving several sewage-treatment plants were found to be between 0.17 and 59.2 µg l$^{-1}$. Moreover, ibuprofen concentration reported in effluents in France and Sweden were 7.11 and 85 µg l$^{-1}$, respectively [8]. The IBU concentration varies and depends on the geographically polluted regions, and contaminated environment (stream, river, etc.). For example, the water column of Lake Greifensee (Switzerland) was mean 1.3 µg l$^{-1}$ for ibuprofen [9], while this number in the Höje River, Sweden, was from 0.12 to 2.2 µg l$^{-1}$ [10].

It is extensively used as non-prescription medicine, with an annual consumption of several hundreds of tons in developed countries. For example, in France, UK and Spain, it was reported that the volume of pharmaceutically active compounds sold in different countries was great, for ibuprofen, at more than 240, 330 and 276 tons only in 2004 [8]. Moreover, the excretion rate of ibuprofen is high (up to 8%) with an incomplete metabolite, probably leading to the penetration of ibuprofen into soil, aquatic media, even human's food source; therefore, it is important to eliminate IBU from water better than other pharmaceutic contaminants [8].

Several recent technologies including membrane distillation, adsorption, advanced oxidation processes (AOPs) and electrochemical oxidation have been developed to eliminate the ibuprofen compound from water [11–14]. For example, Méndez-Arriaga et al. [15] used ultrasonic waves as a means of treatment for the degradation of water contaminated with ibuprofen and obtained the promising results, at 98% within 30 min. Meanwhile, Ali et al. [16] reported the green synthesis of a composite nanoscaled-iron as new generation adsorbent for 92% removal of ibuprofen upon natural water resource conditions (pH 7, low iron dose and agitation time). However, adsorption using porous carbons is demonstrated as the most favourable pathway to treat a wide range of organic compounds consisting of pharmaceuticals [17]. However, finding and designing the robust, efficient, recyclable adsorbents compatible for treatment performance has been still a challenge.

The metal-organic frameworks (MOFs) belong to the crystalline porous materials, assembled by metal clusters and organic linkers [18–20]. With their excellent tailorability and versatile functionalities, the MOFs are applicable as promising catalysts, adsorbents and drug delivery systems [18–23]. Recently, Fe-based MOFs have been used as promising self-sacrificed templates for in situ preparation of hierarchically porous carbon via pyrolysis under aerobic conditions [24]. Generally, the mesoporous carbon (MPC) can be synthesized by a two-step procedure. The first stage focuses on assembling two components including iron salts and carboxylic acid ligands to generate the Fe-MOFs precursor. The second can be followed by pyrolysis of Fe-MOFs employed as templates beneath the inert atmosphere.

Techniques for the preparation of MPC nanomaterials obtain several obvious advantages. The organic ligands constructing the structure of Fe-MOFs are rapidly decomposed under high temperature, then providing a carbonaceous matrix, which plays a role in iron (III) 'in situ reduction' [25,26]. Porous carbon coatings may also increase the surface area and improve functionalized surface chemistry, facilitating the contact between adsorbent and adsorbate [27–29]. Moreover, because abundant Fe−O coordination clusters are order-distributed on crystalline networks or secondary building units (SBUs), as-synthesized nanoscale zero-valent iron (nZVI) particles are well dispersed during the pyrolysis, leading to a higher content, controlled pore sizes and uniform distribution of nZVI encapsulated by carbon [30]. Therefore, the nanostructured materials derived from Fe-MOFs bring a series of catalytic applications [31]. For example, Santos et al. [32] designed well-dispersed

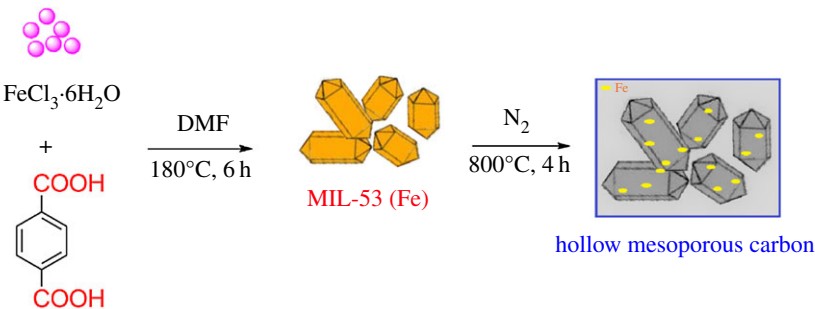

**Scheme 1.** Schematic illustration for the synthesis of the MIL-53 (Fe) and MPC.

nZVI imprinted in the porous carbon matrix from Fe(BTC) (BTC = benzene-1,3,5-tricarboxylic acid) pyrolysis with the amazing Fe loadings (up to 77%) essential for Fischer–Tropsch reactions.

As inspired, we widened the applications of MOFs in various fields [31,33,34]. Herein, MPC nanostructure was directly transformed from Fe-based MOFs precursor named MIL-53 (Fe) using the pyrolysis technique, which occurred at 800°C under a nitrogen atmosphere. The material was then characterized using several measurements and analysis techniques, such as X-ray diffraction (XRD), scanning electron microscopy (SEM), transmission electron microscopy (TEM) and Brunauer–Emmett–Teller (BET). The adsorption experiments of ibuprofen pharmaceutical were conducted to have insight into the effects of concentration, contact time, dosage, pH solution and recyclability. To our best knowledge, this is the first time that the magnetically and hierarchically MPC from MOF MIL-53 (Fe) was adopted for the treatment of ibuprofen drug.

# 2. Experimental procedure

## 2.1. Chemicals and instruments

Chemicals and instruments for the synthesis and characterization of MIL-53 (Fe) and MPC materials were described in electronic supplementary material. In addition, adsorption kinetic, isotherm equations and mathematical formula were addressed.

## 2.2. Preparation of MIL-53 (Fe) and MPC materials

The MIL-53 (Fe) precursor could be facilely synthesized by the solvothermal strategy. Firstly, 1.35 g of $FeCl_3 \cdot 6H_2O$ and 0.83 g of terephthalic acid were dissolved in 25 ml N,N-dimethylformamide (DMF). The mixture was then transferred into a Teflon-lined autoclave and heated up at 180°C for 6 h. The solid was extracted, washed with $C_2H_5OH$ three times (3 × 10 ml) and dried at 110°C.

The MPC was fabricated using a pyrolysis system [35]. Firstly, MIL-53 (Fe) precursor was carefully loaded on a heat-resistant vessel connected with a tube furnace and pyrolysed at 800°C for 4 h under $N_2$ (100 cm³ min⁻¹). The sample was cooled overnight and stored in a desiccator cabinet. Scheme 1 gives an overall picture of the preparation process of MIL-53 (Fe) and hollow MPC.

## 2.3. Experimental batches

Herein, the MPC (0.1 g l⁻¹) was mixed with 50 ml of ibuprofen solutions (10 mg l⁻¹), which were diluted from a stock solution (20 mg l⁻¹). The test tubes were sealed and placed in the shaking tables (200 r.p.m.). After the regular time intervals (30, 60, 120, 240 and 360 min), sample concentrations were analysed using UV–vis spectroscopy at 222 nm. Regarding adsorption isotherms, the similar procedure was employed at various ibuprofen concentrations (5, 10, 15 and 20 mg l⁻¹) at the equilibrium of 240 min. The percentage of removal $H$ (%) and adsorption capacity $q$ (mg g⁻¹) were calculated by the following equations:

$$H(\%) = \frac{C_o - C_e}{C_o}.100, \tag{2.1}$$

$$q_t = \frac{C_o - C_t}{m}.V \tag{2.2}$$

and

$$q_e = \frac{C_o - C_e}{m}.V, \tag{2.3}$$

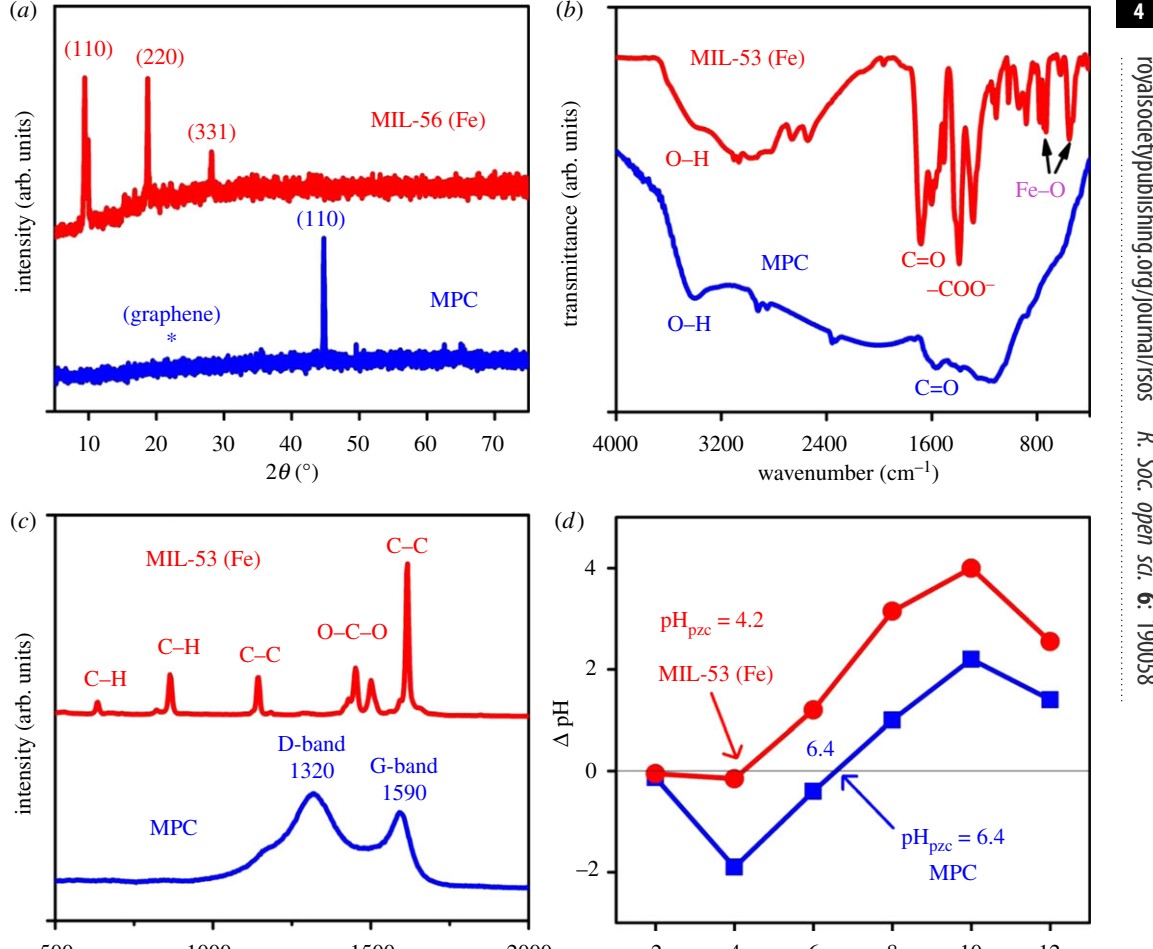

**Figure 1.** (*a*) The XRD, (*b*) FTIR, (*c*) Raman and (*d*) pH$_{pzc}$ profiles of MIL-53 (Fe) and MPC materials.

where $C_o$, $C_t$ and $C_e$ are initial, time $t$ (min) and equilibrium concentrations (mg l$^{-1}$), respectively; $m$ (g) and $V$ (ml) are the amount of adsorbent and volume of solution, respectively.

## 2.4. Determination of pH$_{pzc}$ (pH point of zero charges)

The steps of pH$_{pzc}$ determination were carried out similarly to a recent report [36]. Firstly, the solutions of potassium chloride (KCl) 0.1 M were prepared, and then adjusted with 'initial pH' points (2, 4, 6, 8, 10, 12). An amount of 5.0 mg materials was added into each 25 ml of KCl solution. The mixtures were shaken slightly for 10 min, and maintained stably within 24 h. To identify the 'final pH', the solids were extracted from the solution using a simple magnet. A graph of 'initial pH' against 'final pH' was plotted to visualize the pH$_{pzc}$.

# 3. Results and discussion

## 3.1. Textual characterization

Herein, figure 1*a* compares the X-ray diffraction profiles of MIL-53 (Fe) precursor and MPC materials. The powder XRD patterns (main peaks at around 9.4°, 19° and 28°) of the synthesized MIL-53 (Fe) sample were in line with a previous report, indicating that MIL-53 (Fe) was successfully fabricated [37]. Meanwhile, the crystalline profile for MPC provided clear evidence of the existence of both nZVI portions (JCPDS 87–0721) at around 44.5° (110), 65.0° (220) and an infinitesimal amount of iron oxides crystalline phases at around 35.4° (331). Additionally, the presence of graphitic carbon can be confirmed by broad diffraction from 20° to 30° [36]. The formation of graphitic carbon may be due to

**Table 1.** Surface groups (mmol g$^{-1}$) obtained from Boehm titrations and textual properties of MIL-53 (Fe) and MPC.

| no. | materials | MIL-53 (Fe) | MPC |
|---|---|---|---|
| 1 | carboxylic groups (mmol g$^{-1}$) | 0 | 1.05 |
| 2 | lactonic groups (mmol g$^{-1}$) | 0 | 0.5 |
| 3 | phenolic groups (mmol g$^{-1}$) | 0 | 0.65 |
| 4 | total oxygenated groups (mmol g$^{-1}$) | 0 | 2.2 |
| 5 | total basic groups (mmol g$^{-1}$) | 0 | 0.85 |
| 6 | $S_{BET}$ (m$^2$ g$^{-1}$) | 7.6 | 199.0 |
| 7 | magnetization saturation (emu g$^{-1}$) | 0 | 6.3 |

the direct carbonization of MIL-53 (Fe) at 800°C, converting the carboxylate linkers (H$_2$BDC) into graphitic carbon. The presence of this reductive carbon may stimulate *in situ* chemical reduction (ISCR) to transform Fe (III) species to nZVI nanoparticles [24].

The surface chemistry involving functional groups, which are essential for adsorption, can be analysed using the Fourier transform infrared (FTIR) spectra [38]. According to recorded profiles in figure 1b, the common functional groups of both MIL-53 (Fe) and MPC were both detected at around 3340 cm$^{-1}$ (O–H groups), 1594 cm$^{-1}$ (C = O groups) and 870 cm$^{-1}$ (aromatic C–H) [35]. Moreover, table 1 also reveals the number of functional groups including total oxygenated (2.2 mmol g$^{-1}$) and basic (0.85 mmol g$^{-1}$) groups for MPC via Boehm titration. Importantly, the absence of respective vibrations at around 733 and 550 cm$^{-1}$ (Fe–O) [39,40] on the MPC demonstrated that the cracking of Fe–O coordination bonds on the MIL-53 (Fe) occurred successfully. This observation is totally commensurate with several recent reports, in which the formation of nZVI was attributable to the reduction in Fe(III) on Fe-based MOFs by graphitic carbon under high temperature [32,41–43].

Raman spectra of MIL-53 (Fe) and MPC are revealed in figure 1c. As observed from figure 1c, the appearance of the shifts at around 1456 and 1613 cm$^{-1}$ is characterized by COO– and aromatic C=C groups, respectively [44]. Meanwhile, in the MPC structure emerged the typical D- (1320 cm$^{-1}$) and G- (1590 cm$^{-1}$) bands, indicating the defective structural phase, disorder of MPC [26]. Meanwhile, figure 1d discloses the diagnostic plots of pH$_{pzc}$—one of the very crucial parameters in adsorption, which determine the nature of the surface of a dispersed solid phase at a solid–electrolyte solution interface [45,46]. Herein, the pH$_{pzc}$ values of MIL-53 (Fe) and MPC were 4.2 and 6.4, respectively.

To gain insight into the surface chemical compositions and chemical states, X-ray photoelectron spectroscopy (XPS) analysis was performed. Initially, the XPS survey spectra display that both MIL-53 (Fe) and MPC surfaces are constituted by C, O and Fe elements as shown in figure 2a. According to figure 2b, the typical photoelectron peaks at around 710.6 and 723.9 eV represent the respective sublevels of Fe 2p$_{3/2}$ and Fe 2p$_{1/2}$ for both materials. However, a considerable increase (approx. 30%) in Fe$^{2+}$/Fe$^{3+}$ ratio in MPC compared with MIL-53 (Fe) (electronic supplementary material, table S2) implies that Fe$^{3+}$ in MIL-53 (Fe) can be reduced to lower oxidation states during the pyrolysis of MIL-53 (Fe). Therefore, coexistence of Fe$^{2+}$ and Fe$^{3+}$ species in a mixture on the nZVI surface is highly possible, regarding binding energies of 708.7, 709.7, 710.5, 711.2, 712.0 and 713.2 eV (see electronic supplementary material, table S2) [47–49]. As addressed from XRD and FTIR spectra, proofs of the existence of nZVI embedded in carbon were explored. However, typical XPS signals for nZVI at 706 eV were not observed herein because X-ray photoelectron sensibility merely explores a limited depth (less than 10 nm), suggesting that the reduction in Fe$^{3+}$ species in MIL-53 (Fe) incompletely occurred and these iron oxides may encapsulate the surface of core-shaped nZVI nanoparticles [26,48,50].

Moreover, the O 1s XPS spectra in figure 2c and quantity results obtained from electronic supplementary material, table S2 reveal the dramatic change in the amount of carbonyl groups from 85.4% in MIL-53 (Fe) to 23.1% in MPC. This evidence again demonstrates the strong deconstruction of carboxylate ligands under high temperature to form various kinds of oxygenated groups such as chemisorbed O/O–C=O, C–O, C=O and iron oxides Fe–O, corresponding to binding energies 535.2, 533.5, 532.8 and 530.2 eV, respectively [51]. Meanwhile, the C 1s XPS profile in figure 2d indicates the presence of π−π interaction/O–C=O 289.5 (eV), C–O (286.1 eV), C=O (288.3 eV) and C–C/C=C (284.4 eV) [51]. Remarkably, the ratio of totally non-oxygenated C to oxidized C decreased from 54.8% (MIL-53 (Fe)) to 46.1% (MPC) (electronic supplementary material, table S2). This observation can be explained due to the participation of non-oxygenated C as reductive agent in ISCR process [24].

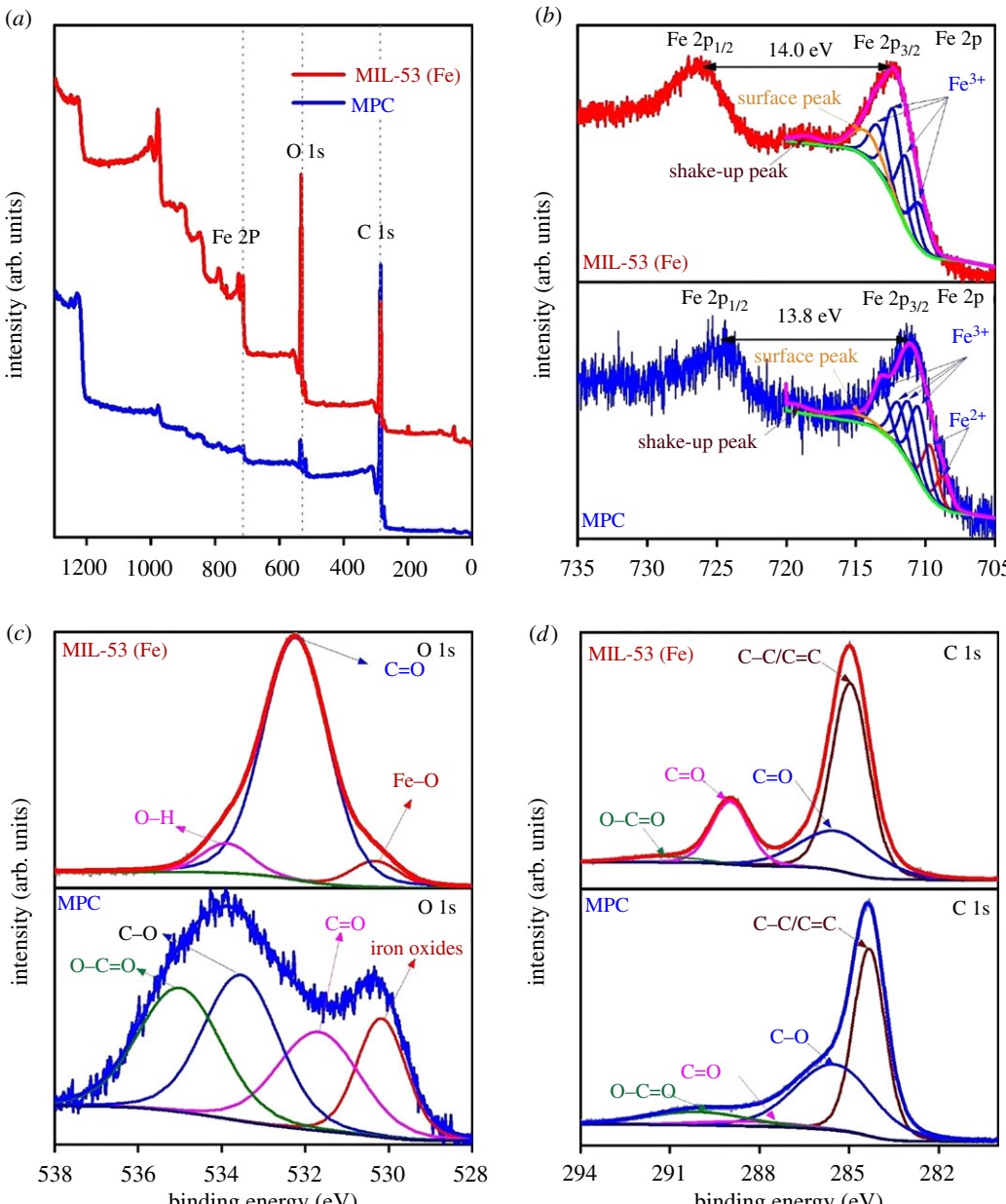

**Figure 2.** The XPS spectra of MIL-53 (Fe) and MPC: (a) survey, (b) Fe 2p, (c) O 1s and (d) C 1s.

The morphological properties by SEM technique were recorded to characterize the structure of the synthesized MPC material and its precursor MIL-53 (Fe). Figure 3a–c displays the polyhedron well-shaped crystalline structure and a uniformly smooth surface of MIL-53 (Fe), in tune with the scrutiny of a recent report in MIL-53 (Fe) [37]. Meanwhile, figure 3d–f discloses the relatively amorphous and defective structure of MPC. The structural observation is consolidated by TEM analysis in figure 3g,h. TEM image in figure 3g shows a consistent structure of MIL-53 (Fe), while intrinsic structure of MPC exposed distinguishable dark spots (Fe nanoparticles inside) covered by opaque regions (graphitized carbon outside) (figure 3h). Because Fe clusters in MIL-53 (Fe) account for construction of crystals through SBUs; therefore, iron distribution in SBUs is entirely homogeneous (figure 3g). However, the collapse of MIL-53 (Fe) structure under high temperature can lead to the rearrangement of iron components. The dispersion of nZVI nanoparticles in carbon again proved that Fe (III) species in SBUs were *in situ* reduced to nZVI via ISCR during pyrolysis of MIL-53 (Fe), then followed by aggregation of Fe nanoparticles under the magnetic effect [31,32,41]. Interestingly, nZVI nanoparticles still exhibit the core–shell structure with 10–20 nm in diameter. Combined with SEM and TEM analysis techniques, nZVI nanoparticles were successfully embedded in the carbonaceous structure.

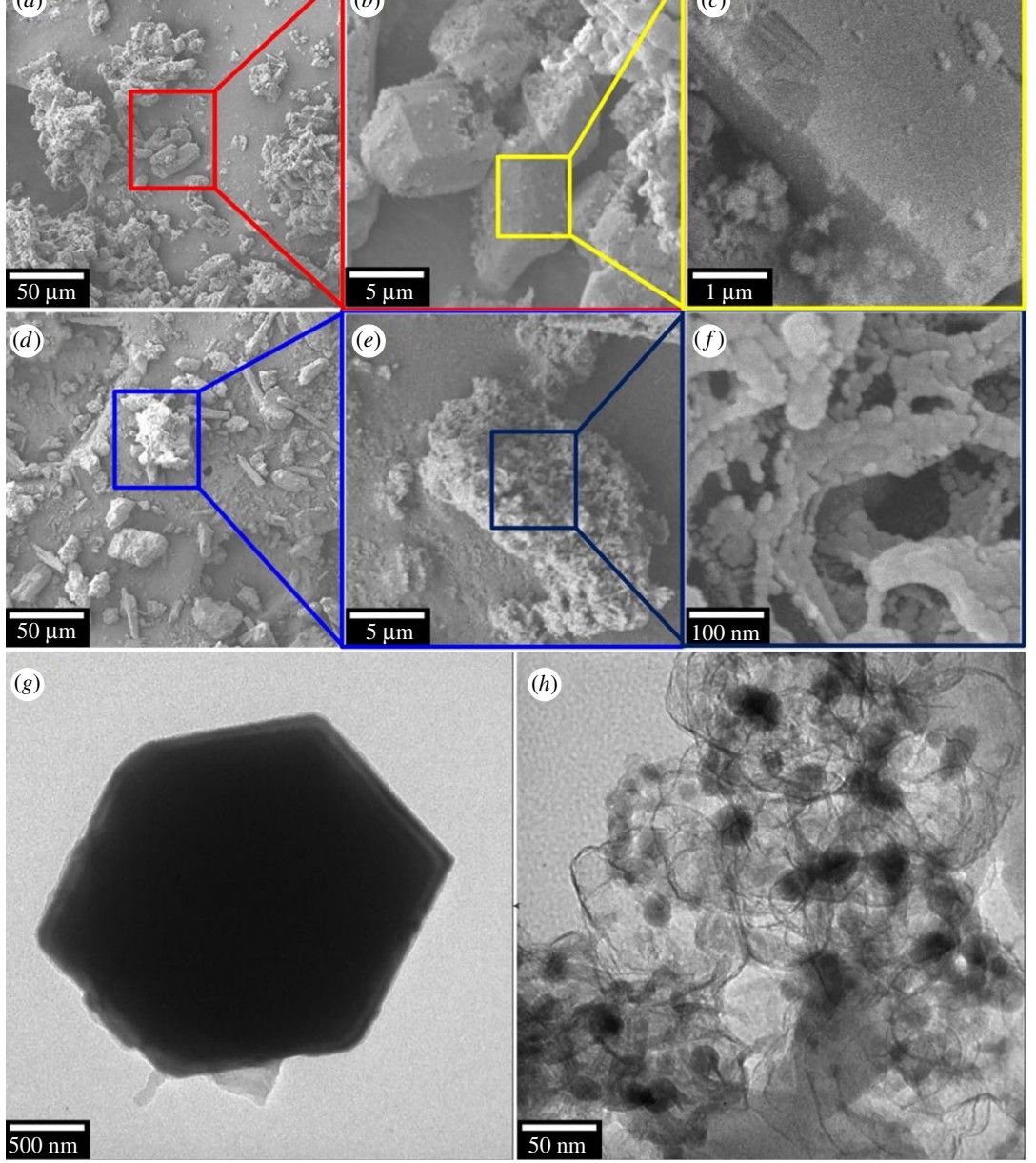

**Figure 3.** ($a-f$) The SEM and ($g,h$) TEM images of MIL-53 (Fe) ($a-c,g$) and MPC ($d-f,h$) materials.

Electronic supplementary material, figure S2 shows the nitrogen adsorption/desorption isotherm and pore distribution curves of MIL-53 (Fe) and MPC. Electronic supplementary material, figure S2a,b demonstrates the dominant presence of mesopores by hysteresis loops obtained from isotherm plots of MIL-53 (Fe) and MPC. Meanwhile, table 1 indicates that the BET surface area and pore volume of MIL-53 (Fe) were $7.6\,m^2\,g^{-1}$ and $0.0118\,cm^3\,g^{-1}$, respectively, while these values for MPC were $199.0\,m^2\,g^{-1}$ and $0.45\,cm^3\,g^{-1}$. Specially, the pore size of MPC was measured at 13.9 Å, which is larger than ibuprofen molecular size (4.3–10.6 Å) (electronic supplementary material, figure S1). The very low surface area of MIL-53 (Fe) can be interpreted due to its inaccessible pores and 'breathing effect' [37]. Because of the higher surface area and larger pore volume parameters, the MPC may generate novel properties applicable for adsorption of ibuprofen.

## 3.2. Adsorption experiments

### 3.2.1. Effect of MPC dosage on ibuprofen adsorption

Optimizing the adsorbent dosage was performed by varying the amount of MPC ($0.025–0.2\,g\,l^{-1}$) added into the ibuprofen solution $10\,mg\,l^{-1}$ at pH 3. As observed from figure 4a, adsorption capacities of

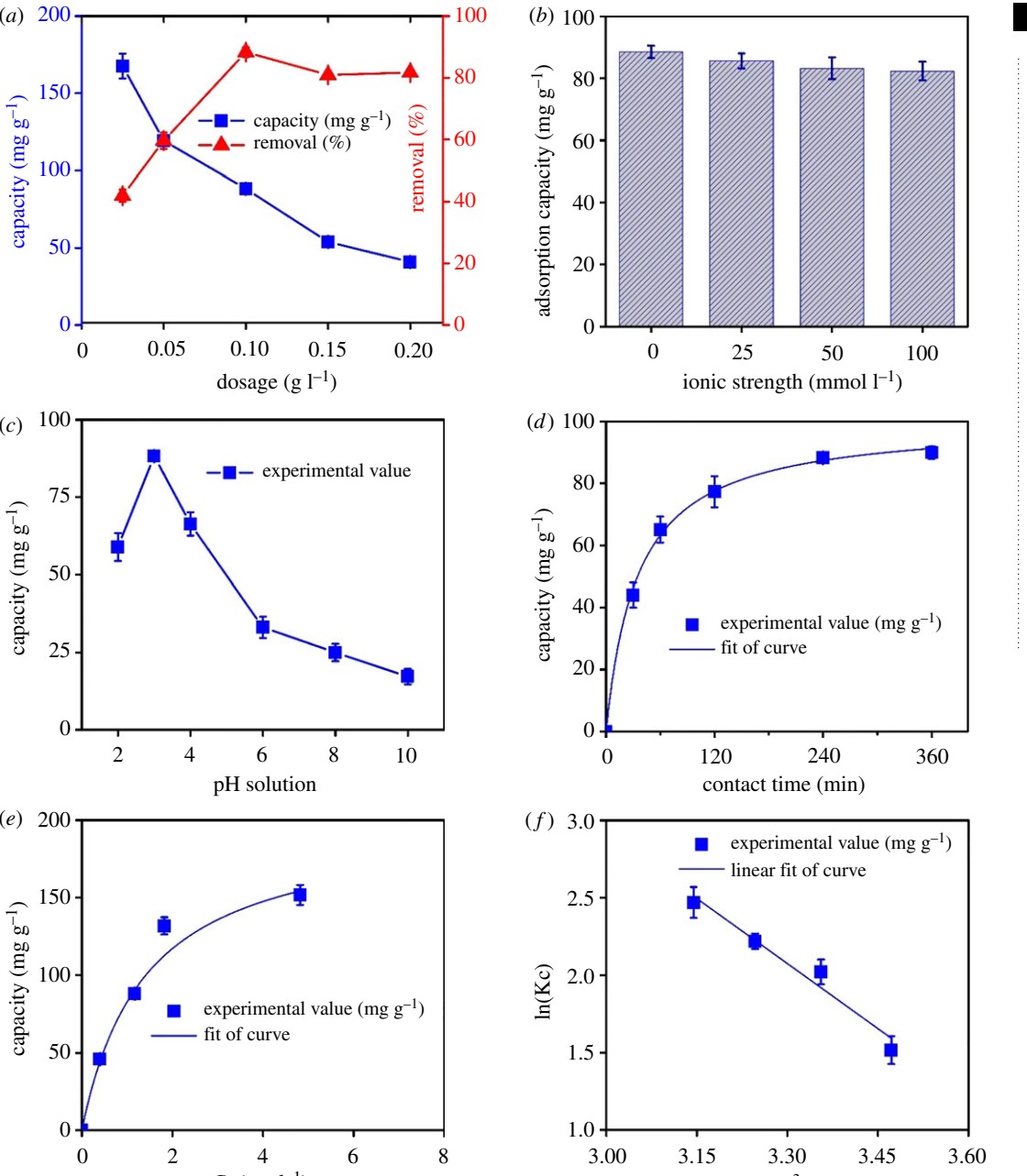

**Figure 4.** (a) Effect of dosage, (b) ionic strength, (c) pH solution, (d) contact time, (e) concentration and (f) temperature on the adsorption of ibuprofen onto MPC material.

ibuprofen gradually decreased with the increased amount of MPC material. For example, nearly 170.0 mg of ibuprofen was adsorbed on per gram of MPC at dosage 0.025 g l$^{-1}$, while that value for dosage 0.2 g l$^{-1}$ was only 40.8 mg. By contrast, removal absorbability of ibuprofen was generally improved with increased dosage of MPC. The optimal dosage value, which ibuprofen removal efficiency reached the peak of 88.3%, was found at 0.1 g l$^{-1}$.

It was reasonable to ascribe the increasing removal percentage of ibuprofen to enlarging the number of active sites by adding a larger quantity of MPC in aqueous solution [3,52,53]. However, the removal of ibuprofen became unconducive at top dosages of MPC, possibly because the considerable addition of MPC may alter the physical properties of solid/liquid suspension (i.e. viscosity), restraining the diffusion of substrate molecules on the MPC surface [54]. Therefore, to minimize the quantity of MPC used, we carried out experiments with adsorbent dosage of 0.1 g l$^{-1}$ for the decontamination of ibuprofen in water.

**Table 2.** Kinetic constants for the adsorption of ibuprofen on MPC.

| kinetic models | parameters | unit | value |
|---|---|---|---|
| pseudo-first-order | $k_1$ | $min^{-1}/(mg\ l^{-1})^{1/n}$ | 0.0154 |
| | $q_e$ | $mg\ g^{-1}$ | 70.74 |
| | $R^2$ | | 0.9946 |
| pseudo-second-order | $k_2$ | $g\ (mg\ min)^{-1} \times 10^4$ | 2.97 |
| | $q_e$ | $mg\ g^{-1}$ | 99.21 |
| | $H = k_2 q_e^2$ | | 2.926 |
| | $R^2$ | | 0.9994 |
| Elovich | $\beta$ | $g\ mg^{-1}$ | 0.0236 |
| | $\alpha$ | $mg\ (g\ min)^{-1}$ | 30.38 |
| | $R^2$ | | 0.9543 |
| Bangham | $k_B$ | $ml\ (g\ l^{-1})^{-1}$ | 0.0429 |
| | $\alpha_B$ | | 0.5528 |
| | $R^2$ | | 0.9689 |

### 3.2.2. Effect of ionic strength on ibuprofen adsorption

The existence of inorganic salts (i.e. NaCl) has an enormous impact on the absorbability of adsorbents because they are likely to vary the solubility of adsorbate and electrostatic interaction between adsorbent and adsorbate [55]. Therefore, to assess the impact of ionic strength on ibuprofen absorbability on MPC, adsorption process was carried out in the presence of $Na^+$ cations at various concentrations from 0.0 to 100.0 mmol $l^{-1}$ and the adsorption capacities measured are shown in figure 4$b$.

A slight decline in the amount of ibuprofen adsorbed on MPC was observed when NaCl concentrations rose. The similar trends have also been reported in the recent literature [2,56]. Theoretically, the cationic state of ibuprofen ($pK_a = 4.9$) is predominant in acidic media ($pH < pK_a$) and net surface charge of MPC is wholly positive at $pH = 3$. A rise in $Na^+$ concentration could deplete the adsorbability of neutral species (i.e. ibuprofen). This may be due to the migration of $Na^+$ cations to the active sites of adsorbent. Tan *et al.* [57] reported the inorganic cations might compete with cationic species on the surface, resulting in a rapid decrease in chemical affinity between the ionic species and the adsorbents. As constructed by magnetic particles, an increase in ionic strength can also lead to an enhancement in particle aggregation of MPC, which reduces the sorption of ibuprofen [55,58].

### 3.2.3. Effect of pH on ibuprofen absorbability

The formation of surface charge on adsorbent and ionization of adsorbate in aqueous solution is gradually influenced by pH values because they control the electrostatic interaction between the adsorbent and the adsorbate [59]. Therefore, we explored the effect of pH solution in the range of 2–10 on the absorbability of ibuprofen on MPC nanocomposite. Note that the acidity, neutrality and basicity of ibuprofen solution can facilely be adjusted by NaOH and HCl solutions, and $pH_{pzc}$ value for MPC was herein measured at 6.5.

As shown in figure 4$c$, the most favourable adsorption of ibuprofen occurred in acidic solutions, which reached the peak of 88.5 mg $g^{-1}$ at the optimal pH 3. At very low pH values (i.e. pH 2), the adsorption efficiency was considerably unconducive. These observations are also commensurate with several recent reports on the adsorption of ibuprofen using various materials [1,51,55].

Typically, as pH values work out smaller than $pK_a$ 5.0, the MPC surface is positively charged, and ibuprofen molecules are present under their cationic form; thus, electrostatic repulsion between two positively charged objects decreases the adsorption capacity. By contrast, if the pH value is higher than $pK_a$ (ibuprofen) but lower than the $pH_{pzc}$ value of MPC, electrostatic attraction between ibuprofen anions and the positively charged surface of MPC is formed to improve the adsorption

capacity. The adsorption capacity observed at pH 3 is noticeably greater than that at pH 5.0–6.0; therefore, electrostatic interaction is ineligible to spell out the dominance of adsorption capacity. Guedidi et al. [51] indicated the dispersive interactions might significantly contribute to the adsorption mechanism of ibuprofen in strongly acidic solutions.

Generally, lone-pair electrons of oxygen atoms electronically interact with protons via a dipole moment effect. Therefore, the attendance of electron-rich functional groups on the adsorbent surface is necessary. Herein, according to FTIR spectra, XPS characterization and Boehm titrations, MPC surface chemistry contains functional groups such as phenolic, lactonic and carboxylic groups, which can provide H-donors to form 'donor–acceptor' complexes with two H-acceptors of ibuprofen (electronic supplementary material, table S1). Although H-donors of these functional groups are easily deprotonated in the strongly basic solution, they can be protected in the strongly acidic media (i.e. pH 3), thus easily stabilize 'donor–acceptor' complexes in acidic solution. Bhadra et al. [1] also reported the similar observation of the decisive role of donor–acceptor bonds between hydrophobic groups (hydroxyl and phenolic) on MPC for forming the H-bonds.

Whereas the ibuprofen solutions became more basic (pH > 7), the adsorption capacity dramatically decreased, merely $17.3 \, mg \, g^{-1}$ at pH 10. This deficiency is mainly attributable to the electrostatic repulsion between two negatively charged objects including ibuprofen anions and MPC surface [59–61]. Interestingly, regardless of experimental conditions at the various pH ranges, the adsorption process of ibuprofen on the MPC still progressed. In fact, there are vital factors playing their roles in maintaining the adsorption equilibrium at even harsh pH values such as $\pi-\pi$ interaction, in which $\pi$-electrons of aromatic rings (MPC) interact with lone-pair electrons of the functional group (ibuprofen). Moreover, another different force such as Van der Waals may also enable the formations of dipole moments [62].

### 3.2.4. Adsorption kinetics

The effect of contact time on the adsorption of ibuprofen over MPC was investigated from 0 to 360 min. According to figure 4d, the ibuprofen adsorption capacity over MPC rapidly boosted in the first 60 min, but steadily increased for the next 180 min. Finally, the equilibrium time was obtained after 240 min. The pattern for this kind of adsorption was totally in line with several reports on the sorption of ibuprofen [1,16,51,58].

The adsorption kinetics of ibuprofen over MPC was studied using four models including pseudo-first-order and pseudo-second-order, Elovich and Bangham equations (electronic supplementary material, equations S1–S5), whose kinetic parameters are displayed in table 2 and linear fitness of curves are plotted in electronic supplementary material, figure S3. As seen from table 2, the coefficient of determination ($R^2$) for all regression models was very high (0.9543–0.9994), suggesting the closeness of predicted data to the observed data. In a previous publication, Ali et al. [16] also reported adsorption of ibuprofen on iron nanoparticles Fe (0) from the aqueous phase well obeyed the mentioned adsorption kinetics.

Electronic supplementary material, figure S3 and table 2 show that the pseudo-second-order model was the most suitable, demonstrated by the extremely high coefficient of determination ($R^2 = 0.9994$) of the linear plot. Therefore, adsorption of ibuprofen over MPC can be chemisorption via electrostatic attraction between adsorbent and adsorbate [63]. Unlike the adsorption behaviour described by pseudo-first-order model, which reflects the rate of adsorption relating to the number of unabsorbed sites, chemisorption generally occurred through rate-controlling steps and diffusion mechanism, and is influenced by functional groups on the surface [64]. Note that chemisorption is characterized by the interaction of chemical groups between adsorbent and adsorbate. It is understandable that the more chemical functional groups (acidic, lactonic, phenolic, basic groups) exist on the surface of MPC, the better the adsorption of ibuprofen is facilitated to occur. In fact, we determined the quality and quantity of functional groups on the surface of MIL-53 (Fe) and MPC via Boehm titration. According to table 1, it is revealed that the number of functional groups include total oxygenated ($2.2 \, mmol \, g^{-1}$) and basic ($0.85 \, mmol \, g^{-1}$) groups for MPC. The above functional groups may contribute to enhancing the adsorbability of MPC towards ibuprofen. Bui & Choi [65] also demonstrated that surface functional groups are a key factor for adsorption of ibuprofen. Moreover, there are many works that proved the adsorption of ibuprofen and other drugs onto MOFs-derived MPC was the chemisorption process with the crucial role of functional groups [1,66–69]. Consequently, we argue that the chemisorption may be a dominance of ibuprofen adsorption in this study.

**Table 3.** Isotherm constants for the adsorption of ibuprofen on MPC.

| isotherm models | parameters | unit | value |
|---|---|---|---|
| Langmuir | $k_L$ | l mg$^{-1}$ | 0.714 |
| | $Q_m$ | mg g$^{-1}$ | 206.5 |
| | $R_L$ | | 0.123 |
| | $R^2$ | | 0.9892 |
| Freundlich | $k_F$ | (mg g$^{-1}$)/(mg l$^{-1}$)$^{1/n}$ | 79.68 |
| | $1/n$ | | 0.4963 |
| | $R^2$ | | 0.9178 |
| Tempkin | $k_T$ | l mg$^{-1}$ | 44.35 |
| | $B_T$ | | 7.41 |
| | $R^2$ | | 0.9436 |
| D–R | $B$ | kJ$^2$ mol$^{-2}$ | 0.12 |
| | $Q_m$ | mg g$^{-1}$ | 142.31 |
| | $E$ | kJ mol$^{-1}$ | 2.0663 |
| | $R^2$ | | 0.94 |

Adsorption and desorption rates were used to simulate the competition between two processes, calculated by electronic supplementary material, equation S4 where $\alpha$ (mg g$^{-1}$ min$^{-1}$) and $\beta$ (g mg$^{-1}$) were adsorption and desorption rate constants, respectively. As extracted from table 2, $\alpha$ and $\beta$ values were 30.38 and 0.0236, respectively, revealing that adsorption outweighed desorption. Moreover, with higher regression constant (0.9689) of Bangham's kinetic model, it is suggested that the intra-particle diffusion mechanism may control the adsorption rate at room temperature [70].

### 3.2.5. Effect of concentration

Adsorption isotherm equations are established to interpret the mechanisms, chemical affinity and surface properties of ibuprofen adsorption over MPC. Firstly, influence of ibuprofen concentration (from 5 to 20 mg l$^{-1}$) on the equilibrium adsorption capacity was studied, and shown in figure 4e. To assess the adsorption isotherms of ibuprofen, experimental data were transformed into various forms to fit with isotherm models including Langmuir, Freundlich, Temkin and Dubinin–Radushkevich (D–R) equations (electronic supplementary material, equations S6–S13), while electronic supplementary material, figure S4 shows the linear regression plots of these isotherm models.

According to table 3, the calculated coefficients of determination $R^2$ were greater than 0.9, revealing the excellent suitability of obtained four models with experimental data. However, based on the $R^2$ values, the compatibility appeared to follow the order: Langmuir > Temkin > D–R > Freundlich. Therefore, the monolayer adsorption might be a dominant mechanism [35]. As calculated from electronic supplementary material, equation S6, the maximum adsorption capacity ($Q_m$) was 206.5 mg g$^{-1}$ (table 3). Moreover, adsorption of ibuprofen drug onto MPC adsorbent is a favourable process because $R_L$ coefficient determined from electronic supplementary material, equation S7 is distributed between 0.025 and 0.521 while $1/n$ coefficient value obtained from electronic supplementary material, equation S8 ranged from 0.1 to 0.5 [71].

To compare the effectiveness in terms of ibuprofen treatment, table 4 summarizes the BET surface area and maximum adsorption capacities of various materials including iron particles and porous carbons. Briefly, with high maximum adsorption capacity in this study compared with other materials, MPC can be an appealing nanocomposite in terms of ibuprofen remediation.

### 3.2.6. Effect of temperature

Figure 4f plots the impact of temperature (288–318 K) on ibuprofen adsorption onto MPC. Thermodynamic constants involving enthalpy ($\Delta H$), entropy ($\Delta S$) and Gibbs free energy ($\Delta G$)

**Table 4.** A comparison of BET surface area and adsorption capacity of adsorbents.

| no. | adsorbents | $S_{BET}$ (m$^2$ g$^{-1}$) | pore volume (cm$^3$ g$^{-1}$) | pore size (Å) | $q_e$ (mg g$^{-1}$) | ref. |
|---|---|---|---|---|---|---|
| 1 | MPC | 199 | 0.39 | 13.9 | 206.5 | this work |
| 2 | AC700N$_2$ | 809 | 0.55 | — | 190.7 | [51] |
| 3 | commercial AC | 800 | 0.52 | — | 160.0 | [51] |
| 4 | H$_2$O$_2$-modified AC | 762 | 0.55 | — | 146.6 | [51] |
| 5 | SBA-15 | 737 | 1.03 | 80 | 0.41 | [65] |
| 6 | cork powder-carbon | 891 | 0.42 | 7.4 | 112.4 | [72] |
| 7 | physically activated cork | 1060 | 0.57 | 11.2 | 378.1 | [72] |
| 8 | physically activated PET | 1426 | 0.584 | 11.0 | 266.6 | [72] |
| 9 | physically activated coal | 1156 | 0.646 | 14.9 | 430.4 | [72] |
| 10 | physically activated wood | 899 | 0.626 | 10.5 | 291.9 | [72] |
| 11 | chemically activated wood | 879 | 0.553 | 11.4 | 149.1 | [72] |
| 12 | CO$_2$-activated carbon | 1055 | 0.733 | 12.0 | 178.0 | [73] |
| 13 | H$_3$PO$_4$-activated carbon | 1106 | 0.560 | 9.0 | 312.7 | [73] |
| 14 | (NH$_4$)$_2$S$_2$O$_8$-activated carbon | 903 | 0.634 | — | 159.8 | [73] |

**Table 5.** Thermodynamic constants for the adsorption of ibuprofen on MPC.

| parameters | unit | value |
|---|---|---|
| $\Delta H°$ | kJ mol$^{-1}$ | $-23.4$ |
| $\Delta S°$ | J mol K$^{-1}$ | 94.5 |
| $\Delta G_{288}$ ($T = 288$ K) | kJ mol$^{-1}$ | $-50.6$ |
| $\Delta G_{298}$ ($T = 298$ K) | kJ mol$^{-1}$ | $-51.6$ |
| $\Delta G_{308}$ ($T = 308$ K) | kJ mol$^{-1}$ | $-52.5$ |
| $\Delta G_{318}$ ($T = 318$ K) | kJ mol$^{-1}$ | $-53.4$ |
| $R^2$ | — | 0.9637 |

are also shown in table 5. An obtained negative $\Delta H$ indicates the adsorption of ibuprofen over MPC was an exothermic process, which totally agreed with recent work [16]. Meanwhile, the positive value of $\Delta S$ shows an increase in disorder occurring in heterogeneous phase because of migration between solvent and ibuprofen molecules during sorption [51]. The negative values of Gibbs free energy from $-50.6$ to $-53.4$ kJ mol$^{-1}$ (table 5) indicated that the adsorption of ibuprofen over MPC was a spontaneous process.

## 3.3. Recyclability study

Reusability study expresses the stability and regeneration of MPC towards decontamination of ibuprofen. Accordingly, eluents are expected to be sustainable and abundant. Recent literature reported that acetone (CH$_3$COCH$_3$) could be used as a green solvent for desorption of ibuprofen from ibuprofen-loaded MPC [1]. Firstly, the solid extracted after the first run was washed with acetone three times ($3 \times 10$ ml), and then was reactivated at 105°C and used for the next reusability study. Figure 5 indicates a negligible decrease (17.5%) from 88.5 mg g$^{-1}$ (1st) to 73.4 mg g$^{-1}$ (5th), suggesting that MPC structure is practically stable to regenerate for many cycles.

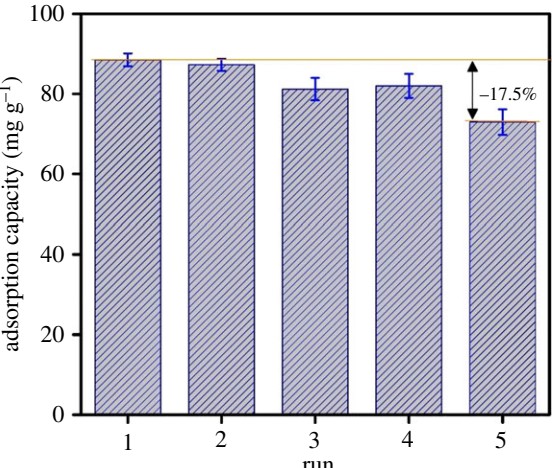

**Figure 5.** Recyclability study of MPC material.

# 4. Conclusion

The MIL-53 (Fe) and MPC materials were successfully fabricated and structurally analysed by several physico-chemical techniques. The characterization results affirmed that the nZVI is entirely embedded in microporous carbon, which obtained the hollow, defective and relatively amorphous structure with the high surface area and large volume. In the adsorption experiments, the pseudo-second-order and Langmuir equations-based $R^2$ coefficients were proved to be the most suitable models to describe the adsorption mechanisms. Moreover, the effects of other parameters were also investigated to reveal the best removal at pH 3, concentration of $10\,\mathrm{mg\,l^{-1}}$, dosage of $0.1\,\mathrm{g\,l^{-1}}$ and time of 4 h. Because of high maximum adsorption capacity and good recyclability, the MPC can be used to remove the ibuprofen from water.

Data accessibility. The datasets supporting this article have been uploaded as part of the electronic supplementary material.

Authors' contributions. T.V.T. and D.T.C.N. conceived and designed the experiments. H.T.N.L., O.T.K.N., V.H.N. and T.T.N. performed the experiments. T.V.T. drafted the first draft of the manuscript, then L.G.B. and T.D.N. corrected the manuscript and co-supervised. T.V.T. directing the research, interpreted and analysed the data, and wrote the full manuscript. All co-authors reviewed the manuscript. All authors gave final approval for publication.

Competing interests. We declare we have no competing interests.

Funding. The study was supported by Science and Technology Incubator Youth Program, managed by the Center for Science and Technology Development, Ho Chi Minh Communist Youth Union, Ho Chi Minh city, Vietnam under the grant no. 06/2018/HĐ-KHCN-VU.

Acknowledgements. The authors gratefully acknowledge the financial support from Center of Science and Technology Development for Youth, Ho Chi Minh City Communist Youth Union, Ho Chi Minh city and many experimental facilities from Nguyen Tat Thanh University, Vietnam for this work.

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
