## [Reviewer comments · Royal Society Open Science]

Review History

RSOS-190058.R0 (Original submission)

Review form: Reviewer 1

Is the manuscript scientifically sound in its present form?

Yes

Are the interpretations and conclusions justified by the results?

Yes

Is the language acceptable?

Yes

Is it clear how to access all supporting data?

Yes

Do you have any ethical concerns with this paper?

No

Have you any concerns about statistical analyses in this paper?

No

Recommendation?

Major revision is needed (please make suggestions in comments)

Comments to the Author(s)

Title: A hollow mesoporous carbon from metal-organic framework for robust absorbability of ibuprofen drug in water

Authors: Thuan Van Tran, Duyen Thi Cam Nguyen, Hanh Thi Ngoc Le, Oanh Thi Kim Nguyen, Vinh Huu Nguyen, Thuong Thi Nguyen, Long Giang Bach, Trinh Duy Nguyen Manuscript No.: RSOS-190058

To author

General comments:

The manuscript entitled "A hollow mesoporous carbon from metal-organic framework for robust absorbability of ibuprofen drug in water" has been reviewed. The authors presented an interesting work about the remediation of ibuprofen by adsorption from a metal-organic framework MIL-53 (Fe). The resulting material was fully characterised using FTIR, XRD, XPS, BET, SEM and TEM. The adsorption experiments were also applied to kinetic and isotherm models to determine the mechanism of adsorption. Recyclability of the material was tested and gave good results. The paper also makes a good contribution to this field of study and presents a suitable material that can be considered for the removal of ibuprofen. However the paper has variety of errors which must be corrected before further consideration for acceptance. Here below are some examples of errors which demand reconsideration:

Specific comments:

1. Some improvements can be made with regard to including instrumentation used, conditions of analysis and any sample preparations carried out - these should be included in the experimental.
2. The novelty of work should be clarified.
3. The PXRD and FT-IR spectra (Fig. 1. a, b) should be indexed, and Zeta potential of samples (Fig. 1. d) at different pHs should be presented.
4. In the section "3.3. Recyclability study", the author said "ibuprofen-MPC complex", the description seems vague. I suggest this text should be better replaced with "ibuprofen-loaded MPC".
5. In the section "3.2.6. Effect of temperature", the thermodynamic study was conducted and Gibbs free energies (ΔG) have been clarified with negative values. Why don't you conclude that the adsorption is a spontaneous process?
6. Several statements need more references, for example "Porous carbon coatings may also increase the surface area and improve functionalized surface chemistry, facilitating the contact between adsorbents and adsorbates" in the background; "733 cm^{-1} and 550 cm^{-1} (Fe-O) on the MPC" in FT-IR analysis section, etc.
7. Why don't the authors study the effect of ibuprofen concentration at higher levels instead of 5-20 mg/L? Please provide more analysis about the effect of contact time in the section "3.2.4. Adsorption kinetics"
8. Many English grammar and spelling errors need to be avoided to ensure that the paper flows and reads well for readers.

Some minor comments:

The authors should also pay attention to the English of the paper to ensure that it flows and reads well. For example:

Page 3, Line 15, 'presented' should be 'presenting'

Page 4, Line 18. I think your presented application is insufficient and I am not very convinced about study objectives. I suggest you to exemplify additional interesting applications of nano Fe-MOF to raise the need for your paper. If possible, give some shortcomings of reviewed papers

and state which will be overcome by your study.

Page 5, Line 44. should be 'against'

Page 6, Line 28. 'which' should be 'in which'

Page 6, Line 39. Please rewrite this sentence

Page 7, Line 12. Please remove 'was'

Page 8, Line 44. I suggested to revise the phrase after '...proportional to...'

Page 9 Line 24. I suggest to rewrite as '...due to competition between cations and active site seizure of inorganic cations'.

Page 9, Line 29. Should be 'reduces'

Page 10, Line 28. Should be 'are protected'. Also, please rewrite the phrase after the comma

Page 10, Line 43. Should be 'playing'

Page 10, Line 46, Should be 'in which' instead of 'which'

Page 10, Line 57, Should be 'was obtained'

Page 11, Line 38. Please write the whole sentence starting from 'moreover,.....'. I suggest splitting it into two separate sentences.

Page 12, Line 21. Should be 'briefly'

Review form: Reviewer 2

Is the manuscript scientifically sound in its present form?

No

Are the interpretations and conclusions justified by the results?

No

Is the language acceptable?

Yes

Is it clear how to access all supporting data?

Yes

Do you have any ethical concerns with this paper?

No

Have you any concerns about statistical analyses in this paper?

No

Recommendation?

Major revision is needed (please make suggestions in comments)

Comments to the Author(s)

It this work, a mesoporous carbon from MIL-53(Fe) is synthetized for the elimination of IBU from water.

Title:

-is it aDsorbability or aBSorbability? May be it need to me modified as the adsorbent is a porous material.

Introduction:

- "Pharmaceutically bioactive compounds (PBCs) are widely consumed over the world

because of their crucial role playing in protecting human's health from the attack of bacteria species". Are the PBCs only used to protect from the attack of bacteria species? what about pain? inflammation? or other diseases?

- The authors mentioned that "With their excellent tailorability and versatile functionalities, the MOFs are applicable as promising catalysts, adsorbents, drug delivery systems 20–24." However, in this references there are no articles related with the drug delivery. They need to complete this information.

- What is the concentration of IBU in wastewater? is it similar to the one used in this study? What is the pH of this wastewater? Why is important to eliminate IBU from water better than other pharmaceutical contaminants?

Experimental

-The authors measured the IBU concentration by UV-vis but at which wavelength??

-Is there any work in bibliography were the elimination of IBU in waterwaste is studied? What it the % of eliminated IBU? For example: "Ultrasonic treatment of water contaminated with ibuprofen". Water Research 42 (16): 4243-4248, 2008. or "Synthesis of composite iron nano adsorbent and removal of ibuprofen drug residue from water" Journal of Molecular Liquids Volume 219, July 2016, Pages 858-864. Among others.

- MIL-53 is used as a source of Fe, as Fe can be used for the "Chemisorption " of IBU. But, how can the authors confirm that is a chemisorption? There is a complete characterization of the MIL-53, and the MPC, but nothing about the IBU loaded MPC. Can the authors add the FTIR, ATG, SEM, BET, DXRP of the loaded matrix?

- In all the data there are no errors, have the authors checked the reproducibility of these results? They have to add the error values and the error bars in all the figures.

Decision letter (RSOS-190058.R0)

25-Feb-2019

Dear Dr D.:

Title: A hollow mesoporous carbon from metal-organic framework for robust absorbability of ibuprofen drug in water

Manuscript ID: RSOS-190058

The editor assigned to your manuscript has now received comments from reviewers. We would like you to revise your paper in accordance with the referee and Subject Editor suggestions which can be found below (not including confidential reports to the Editor). Please note this decision does not guarantee eventual acceptance.

Please submit your revised paper before 20-Mar-2019. Please note that the revision deadline will expire at 00.00am on this date. If we do not hear from you within this time then it will be assumed that the paper has been withdrawn. In exceptional circumstances, extensions may be possible if agreed with the Editorial Office in advance. We do not allow multiple rounds of revision so we urge you to make every effort to fully address all of the comments at this stage. If deemed necessary by the Editors, your manuscript will be sent back to one or more of the original reviewers for assessment. If the original reviewers are not available we may invite new reviewers.

Please also include the following statements alongside the other end statements. As we cannot publish your manuscript without these end statements included, if you feel that a given heading is not relevant to your paper, please nevertheless include the heading and explicitly state that it is not relevant to your work.

- Ethics statement

Please clarify whether you received ethical approval from a local ethics committee to carry out your study. If so please include details of this, including the name of the committee that gave consent in a Research Ethics section after your main text. Please also clarify whether you received informed consent for the participants to participate in the study and state this in your Research Ethics section.

OR

Please clarify whether you obtained the necessary licences and approvals from your institutional animal ethics committee before conducting your research. Please provide details of these licences and approvals in an Animal Ethics section after your main text.

OR

Please clarify whether you obtained the appropriate permissions and licences to conduct the fieldwork detailed in your study. Please provide details of these in your methods section.

On behalf of the Subject Editor Professor Anthony Stace and the Associate Editor Professor Kim Jelfs.

RSC Associate Editor:
Comments to the Author:

(There are no comments.)

RSC Subject Editor:

Comments to the Author:

(There are no comments.)

Reviewers' Comments to Author:

Reviewer: 1

Comments to the Author(s)

Title: A hollow mesoporous carbon from metal-organic framework for robust absorbability of ibuprofen drug in water

Authors: Thuan Van Tran, Duyen Thi Cam Nguyen, Hanh Thi Ngoc Le, Oanh Thi Kim Nguyen, Vinh Huu Nguyen, Thuong Thi Nguyen, Long Giang Bach, Trinh Duy Nguyen Manuscript No.: RSOS-190058

To author

General comments:

The manuscript entitled "A hollow mesoporous carbon from metal-organic framework for robust absorbability of ibuprofen drug in water" has been reviewed. The authors presented an interesting work about the remediation of ibuprofen by adsorption from a metal-organic framework MIL-53 (Fe). The resulting material was fully characterised using FTIR, XRD, XPS, BET, SEM and TEM. The adsorption experiments were also applied to kinetic and isotherm models to determine the mechanism of adsorption. Recyclability of the material was tested and gave good results. The paper also makes a good contribution to this field of study and presents a suitable material that can be considered for the removal of ibuprofen. However the paper has variety of errors which must be corrected before further consideration for acceptance. Here below are some examples of errors which demand reconsideration:

Specific comments:

1. Some improvements can be made with regard to including instrumentation used, conditions of analysis and any sample preparations carried out - these should be included in the experimental.
2. The novelty of work should be clarified.
3. The PXRD and FT-IR spectra (Fig. 1. a, b) should be indexed, and Zeta potential of samples (Fig. 1. d) at different pHs should be presented.
4. In the section "3.3. Recyclability study", the author said "ibuprofen-MPC complex", the description seems vague. I suggest this text should be better replaced with "ibuprofen-loaded MPC".
5. In the section "3.2.6. Effect of temperature", the thermodynamic study was conducted and Gibbs free energies (ΔG) have been clarified with negative values. Why don't you conclude that the adsorption is a spontaneous process?
6. Several statements need more references, for example "Porous carbon coatings may also increase the surface area and improve functionalized surface chemistry, facilitating the contact between adsorbents and adsorbates" in the background; "733 cm^{-1} and 550 cm^{-1} (Fe-O) on the MPC" in FT-IR analysis section, etc.
7. Why don't the authors study the effect of ibuprofen concentration at higher levels instead of 5-20 mg/L? Please provide more analysis about the effect of contact time in the section "3.2.4. Adsorption kinetics"
8. Many English grammar and spelling errors need to be avoided to ensure that the paper flows and reads well for readers.

Some minor comments:

The authors should also pay attention to the English of the paper to ensure that it flows and reads well. For example:

- Page 3, Line 15, 'presented' should be 'presenting'
- Page 4, Line 18. I think your presented application is insufficient and I am not very convinced about study objectives. I suggest you to exemplify additional interesting applications of nano Fe-MOF to raise the need for your paper. If possible, give some shortcomings of reviewed papers and state which will be overcome by your study.
- Page 5, Line 44. should be 'against'
- Page 6, Line 28. 'which' should be 'in which'
- Page 6, Line 39. Please rewrite this sentence
- Page 7, Line 12. Please remove 'was'
- Page 8, Line 44. I suggested to revise the phrase after '...proportional to...'
- Page 9 Line 24. I suggest to rewrite as '...due to competition between cations and active site seizure of inorganic cations'.
- Page 9, Line 29. Should be 'reduces'
- Page 10, Line 28. Should be 'are protected'. Also, please rewrite the phrase after the comma
- Page 10, Line 43. Should be 'playing'
- Page 10, Line 46, Should be 'in which' instead of 'which'
- Page 10, Line 57, Should be 'was obtained'
- Page 11, Line 38. Please write the whole sentence starting from 'moreover,.....'. I suggest splitting it into two separate sentences.
- Page 12, Line 21. Should be 'briefly'

Reviewer: 2

Comments to the Author(s)

It is this work, a mesoporous carbon from MIL-53(Fe) is synthesized for the elimination of IBU from water.

Title:

-is it adsorbability or absorbability? May be it need to be modified as the adsorbent is a porous material.

Introduction:

- "Pharmaceutically bioactive compounds (PBCs) are widely consumed over the world because of their crucial role playing in protecting human's health from the attack of bacteria species". Are the PBCs only used to protect from the attack of bacteria species? what about pain? inflammation? or other diseases?

- The authors mentioned that "With their excellent tailorability and versatile functionalities, the MOFs are applicable as promising catalysts, adsorbents, drug delivery systems 20–24." However, in this references there are no articles related with the drug delivery. They need to complete this information.

- What is the concentration of IBU in wastewater? is it similar to the one used in this study? What is the pH of this wastewater? Why is important to eliminate IBU from water better than other pharmaceutical contaminants?

Experimental

-The authors measured the IBU concentration by UV-vis but at which wavelength??

-Is there any work in bibliography where the elimination of IBU in wastewater is studied? What is the % of eliminated IBU? For example: "Ultrasonic treatment of water contaminated with ibuprofen". Water Research 42 (16): 4243-4248, 2008. or "Synthesis of composite iron nano adsorbent and removal of ibuprofen drug residue from water" Journal of Molecular Liquids Volume 219, July 2016, Pages 858-864. Among others.

- MIL-53 is used as a source of Fe, as Fe can be used for the "Chemisorption" of IBU. But, how can the authors confirm that is a chemisorption? There is a complete characterization of the MIL-53,

and the MPC, but nothing about the IBU loaded MPC. Can the authors add the FTIR, ATG, SEM, BET, DXRP of the loaded matrix?

- In all the data there are no errors, have the authors checked the reproducibility of these results? They have to add the error values and the error bars in all the figures.

Author's Response to Decision Letter for (RSOS-190058.R0)

See Appendix A.

RSOS-190058.R1 (Revision)

Review form: Reviewer 1

Is the manuscript scientifically sound in its present form?

Yes

Are the interpretations and conclusions justified by the results?

Yes

Is the language acceptable?

Yes

Is it clear how to access all supporting data?

Yes

Do you have any ethical concerns with this paper?

No

Have you any concerns about statistical analyses in this paper?

No

Recommendation?

Accept as is

Comments to the Author(s)

The authors have addressed all the comments properly. I think the paper is now publishable.

Review form: Reviewer 2

Is the manuscript scientifically sound in its present form?

Yes

Are the interpretations and conclusions justified by the results?

Yes

Is the language acceptable?

Yes

Is it clear how to access all supporting data?

Yes

Do you have any ethical concerns with this paper?

No

Have you any concerns about statistical analyses in this paper?

No

Recommendation?

Accept as is

Comments to the Author(s)

The authors have successfully respond to all issues indicated in the review report.

Decision letter (RSOS-190058.R1)

15-Apr-2019

Dear Dr D.:

Title: A hollow mesoporous carbon from metal-organic framework for robust absorbability of ibuprofen drug in water

Manuscript ID: RSOS-190058.R1

It is a pleasure to accept your manuscript in its current form for publication in Royal Society Open Science. The chemistry content of Royal Society Open Science is published in collaboration with the Royal Society of Chemistry.

Yours sincerely,

Dr Laura Smith

Publishing Editor, Journals

Royal Society of Chemistry

Thomas Graham House

Science Park, Milton Road

Cambridge, CB4 0WF

Royal Society Open Science - Chemistry Editorial Office

On behalf of the Subject Editor Professor Anthony Stace and the Associate Editor Professor Kim Jelfs.

Reviewer(s)' Comments to Author:

Reviewer: 1

Comments to the Author(s)

The authors have addressed all the comments properly. I think the paper is now publishable.

Reviewer: 2

Comments to the Author(s)

The authors have successfully respond to all issues indicated in the review report.

Appendix A

Dr. Laura Smith,
Publishing Editor, Journals
Royal Society of Chemistry
Thomas Graham House
Science Park, Milton Road
Cambridge, CB4 0WF
Royal Society Open Science - Chemistry Editorial Office

Re: *Revision requested for RSOS-190058*
Manuscript ID: *RSOS-190058*
Title of Paper: *A hollow mesoporous carbon from metal-organic framework for robust adsorbability of ibuprofen drug in water*
Author(s): Thuan Van Tran, Duyen Thi Cam Nguyen, Hanh Thi Ngoc Le, Oanh Thi Kim Nguyen, Vinh Huu Nguyen, Thuong Thi Nguyen, Long Giang Bach, Trinh Duy Nguyen*
Journal: Royal Society Open Science

Dear **Prof. Laura Smith,**

We would like to express our gratitude for the Editor and Reviewer's efforts to improve the quality of our manuscript. We believe that our manuscript as a qualified paper in *Royal Society Open Science*. We have tried our best to respond to all issues indicated in the review report fully. In the revised version, we have highlighted the changes to our manuscript using various colors. Here, we would like to address the reviewer's concerns as follows:

Reviewer #1:

The manuscript entitled "A hollow mesoporous carbon from metal-organic framework for robust adsorbability of ibuprofen drug in water" has been reviewed. The authors presented an interesting work about the remediation of ibuprofen by adsorption from a metal-organic framework MIL-53 (Fe). The resulting material was fully characterised using FTIR, XRD, XPS, BET, SEM and TEM. The adsorption experiments were also applied to kinetic and isotherm models to determine the mechanism of adsorption. Recyclability of the material was tested and gave good results. The paper also makes a good contribution to this field of study and presents a suitable material that can be

considered for the removal of ibuprofen. However, the paper has variety of errors which must be corrected before further consideration for acceptance. Here below are some examples of errors which demand reconsideration:

Specific comments:

1. Some improvements can be made with regard to including instrumentation used, conditions of analysis and any sample preparations carried out - these should be included in the experimental.

We are very thankful for the reviewer's evaluation and suggestion to improve the quality of the manuscript. As mentioned, we clarified this instrumentation, conditions of analysis and sample preparations included in the supporting information (SI) document, as follows: "All chemicals including ibuprofen, 1,4-benzendicarboxylic acid, iron chloride and potassium chloride were commercially purchased from Merck. The D8 Advance Bruker powder diffractometer was used to record the X-ray powder diffraction (XRD) profiles using Cu-K α beams as excitation sources. The S4800 instrument (Japan) was implemented to capture the scanning electron microscope (SEM) images with the magnification of 7000 using an accelerating voltage source (15 kV). The JEOL JEM 1400 instrument was used to study the transmission electron microscopy (TEM). The characteristics of surface chemistry was investigated by the FT-IR spectra on the Nicolet 6700 spectrophotometer. The N₂ adsorption/desorption isotherm and pore size distribution data were recorded on the Micromeritics 2020 volumetric adsorption analyzer system. The UV-Vis spectrophotometer was used to determine the ibuprofen concentration at 222 nm. The photoelectron spectrometer Kratos Axis-Ultra was used to recorded the signals of the X-ray photoelectron spectra (XPS) using a monochromatic X-ray source of Al K α and Casa XPS software was utilized to analyze the XPS spectra. All XPS spectra were calibrated using the C 1s peak (284.8 eV) with a subtraction by Shirley background. Gupta and Sen (GS) multiplets was used to fit the high-spin Fe³⁺ states".

To help the readers follow easily, we inserted these related items in "Section 2. Experimental" (Page 4), and "section 2.1. Chemicals and instruments" which you can track them with **bright green color** as follows:

"Chemicals and instruments for the synthesis and characterization of MIL-53 (Fe) and MPC materials were described in supplementary information (SI) material. In addition, adsorption kinetic, isotherm equations, and mathematical formula were addressed."

2. The novelty of work should be clarified.

The authors highly appreciate the reviewer's comment to emerge the novelty of paper. Herein, we clarify the novelty of paper as follows:

- For the first time, the magnetically and hierarchically mesoporous carbon (MPC) from metal-organic framework MIL-53 (Fe) was adopted for the treatment of ibuprofen drug.
- Characterization, kinetic, isotherm, thermodynamic, and recyclability experiments were systematically studied.
- Adsorption mechanisms including H-bonding, π - π interaction, metal-oxygen, electrostatic attraction were rigorously proposed.
- Compared with several studies, outstanding adsorption capacity (206.5 mg/g), high stability, and good reusability (up to 5 cycles) were reached using MIL-53 (Fe)-based mesoporous carbon.

With points mentioned above, therefore, we are pleased to clarify the novelty in the manuscript as the following texts:

In the abstract, page 2 (lines 11-12), we added: "effects of contact time, MPC dosage, ionic strength, concentration and temperature were systematically investigated".

In the abstract, page 2 (lines 18-20), we added: "Adsorption mechanisms including H-bonding, π - π interaction, metal-oxygen, electrostatic attraction were rigorously proposed".

In the abstract, page 2 (lines 21-24), we modified: "In comparison to several studies, the MPC nanocomposite in this work obtained the outstanding maximum adsorption capacity (206.5 mg/g) and good reusability (5 cycles), thus it can be utilized as a feasible alternative for decontamination of ibuprofen anti-inflammatory drug from water."

In the introduction, page 4 (line 28-30), we added: "To our best knowledge, this is the first time that the magnetically and hierarchically mesoporous carbon from metal-organic framework MIL-53 (Fe) was adopted for the treatment of ibuprofen drug."

3. The PXRD and FT-IR spectra (Fig. 1. a, b) should be indexed, and Zeta potential of samples (Fig. 1. d) at different pHs should be presented.

We highly appreciate the reviewer's comment to make the figure more obvious. According to the suggestion from reviewer, PXRD and FT-IR spectra (Fig. 1. a, b) have been indexed, and Zeta potential of samples (Fig. 1. d) at different pHs has been presented.

In pages 18, we inserted a new version with minor detailed modification as follows:

Fig. 1. The XRD (a), FTIR (b), Raman (c), and pH_{pzc} (d) profiles of MIL-53 (Fe) and MPC materials

4. In the section “3.3. Recyclability study”, the author said “ibuprofen-MPC complex”, the description seems vague. I suggest this text should be better replaced with “ibuprofen-loaded MPC”.

We highly appreciate the reviewer's comment to make the text more obvious. According to the suggestion from reviewer, the authors have adjusted the text “ibuprofen-MPC complex” into “ibuprofen-loaded MPC” under a **bright green color** in page 13.

5. In the section “3.2.6. Effect of temperature”, the thermodynamic study was conducted and Gibbs free energies (ΔG) have been clarified with negative values. Why don't you conclude that the adsorption is a spontaneous process?

We highly appreciate the reviewer's comment. In section 3.2.6. Effect of temperature, we found that the adsorption of ibuprofen onto MPC was an exothermic (negative ΔH) and spontaneous (negative Gibbs free energies ΔG) process. According to the suggestion from reviewer, the authors have had a conclusion about this process.

In page 12, section 3.2.6, we modified a new text under **bright green color** for more analysis, as follows: “The negative values of Gibbs free energy from -50.609 to -53.443 kJ/mol (Table 5) indicated that the adsorption of ibuprofen over MPC was a spontaneous process”.

On the other hand, we adjusted the decimal number of energies in Table 5.

In page 12, line 35, we also adjusted the decimal number of energies “ -50.6 to -53.4 ”.

Table 5. Thermodynamic constants for the adsorption of ibuprofen on MPC

Parameters	Unit	Value
ΔH°	kJ/mol	-23.4
ΔS°	J/mol.K	94.5
ΔG_{288} (T=288 K)	kJ/mol	-50.6
ΔG_{298} (T=298 K)	kJ/mol	-51.6
ΔG_{308} (T=308 K)	kJ/mol	-52.5
ΔG_{318} (T=318 K)	kJ/mol	-53.4
R^2	-	0.9637

6. Several statements need more references, for example “Porous carbon coatings may also increase the surface area and improve functionalized surface chemistry, facilitating the contact between adsorbents and adsorbates” in the background; “733 cm^{-1} and 550 cm^{-1} (Fe–O) on the MPC” in FT-IR analysis section, etc.

We highly appreciate the reviewer’s comment to improve the quality of this manuscript. According to the reviewer’s suggestion, we modified more literatures to support the statements “Porous carbon coatings may also increase the surface area and improve functionalized surface chemistry, facilitating the contact between adsorbents and adsorbates” in the background; and “733 cm^{-1} and 550 cm^{-1} (Fe–O) on the MPC” in FT-IR analysis section, etc.

This is the list of references, which was mentioned in the manuscript revision and appeared under **bright green color**.

1. W. Yao, S. Wu, L. Zhan and Y. Wang, *Chem. Eng. J.*, 2019, 361, 329–341.
2. Q. Hu, M. Yu, J. Liao, Z. Wen and C. Chen, *J. Mater. Chem. A*, 2018, 6, 2365–2370.
3. Q. Gan, H. He, K. Zhao, Z. He and S. Liu, *Electrochim. Acta*, 2018, 266, 254–262
4. T. Van Tran, V. Dai Cao, V. Huu Nguyen, B. N. Hoang, D.-V. N. Vo, T. D. Nguyen and L. Giang Bach, *J. Environ. Chem. Eng.*, 2019, 102902.
5. T. Van Tran, D. T. C. Nguyen, H. T. N. Le, T. T. K. Tu, N. D. Le, K. T. Lim, L. G. Bach and T. D. Nguyen, *J. Environ. Chem. Eng.*, 2019, 7, 102881.

7. Why don’t the authors study the effect of ibuprofen concentration at higher levels instead of 5–20 mg/L? Please provide more analysis about the effect of contact time in the section “3.2.4. Adsorption kinetics”

We highly appreciate the reviewer’s comment to improve the quality of this manuscript. In this study, we conducted the experiments by selecting the appropriate range of ibuprofen concentration from 5–20 mg/L. Performing the adsorption at higher ibuprofen concentration is not referred because the solubility of ibuprofen in water at 25 °C is found at approximately 21 mg/L [1]. Sung et al. also

reported that ibuprofen concentration in their work limited to 21 mg/L due to low solubility of ibuprofen [2]. To obtain the higher concentration, it may be under the use of surfactants such as polyvinyl pyrrolidone (PVP), sodium lauryl sulphate (SLS) or other methods according to recent studies [3–5]. However, we solely carried out the experiments in water without surfactants, therefore, it is reported to be 5 – 20 mg/L for ibuprofen concentration.

For more analysis about the effect of contact time in the section “3.2.4. Adsorption kinetics”, the authors modified the new text **bright green color**.

In page 11, a paragraph has been inserted as follows:

“The effect of contact time on the adsorption of ibuprofen over MPC was investigated from 0 to 360 minutes. According to the Fig. 4 (d), the ibuprofen adsorption capacity over MPC rapidly boosted in the first 60 min, but steadily increased for the next 180 min. Finally, the equilibrium time obtained after 240 minutes. The pattern for this kind of adsorption was totally in line with several reports on the sorption of ibuprofen [2,6–8]”.

8. Many English grammar and spelling errors need to be avoided to ensure that the paper flows and reads well for readers.

Some minor comments:

The authors should also pay attention to the English of the paper to ensure that it flows and reads well. For example:

Page 3, Line 15, ‘presented’ should be ‘presenting’

Page 4, Line 18. I think your presented application is insufficient and I am not very convinced about study objectives. I suggest you to exemplify additional interesting applications of nano Fe-MOF to raise the need for your paper. If possible, give some shortcomings of reviewed papers and state which will be overcome by your study.

Page 5, Line 44. should be ‘against’

Page 6, Line 28. ‘which’ should be ‘in which’

Page 6, Line 39. Please rewrite this sentence

Page 7, Line 12. Please remove 'was'

Page 8, Line 44. I suggested to revise the phrase after '...proportional to...'

Page 9 Line 24. I suggest to rewrite as '...due to competition between cations and active site seizure of inorganic cations'.

Page 9, Line 29. Should be 'reduces'

Page 10, Line 28. Should be 'are protected'. Also, please rewrite the phrase after the comma

Page 10, Line 43. Should be 'playing'

Page 10, Line 46, Should be 'in which' instead of 'which'

Page 10, Line 57, Should be 'was obtained'

Page 11, Line 38. Please write the whole sentence starting from 'moreover,.....'. I suggest splitting it into two separate sentences.

Page 12, Line 21. Should be 'briefly'

The authors highly appreciate the reviewer's corrections to improve the quality of paper. We completely agree with the reviewer's suggestion, and for these concerns, we are pleased to correct these English grammar errors.

Page 3, Line 15, 'presented' has been transformed into 'presenting'

Page 4, Line 18. The authors attempted to refer to several works concerning about the application of nanostructured MOFs. Therefore, we cited these shortcoming references in our manuscript.

Page 5, Line 44. We corrected this grammar: "agaisnt" into "against"

Page 6, Line 28. We corrected this grammar: "which" into "in which"

Page 6, Line 39. We rewrote this sentence.

"Fig. 1 (d) discloses the diagnostic plots of pH_{pzc} , which the surface charge components reach neutral under given conditions of temperature, applied pressure and aqueous solution composition"

Into

“Fig. 1 (d) discloses the diagnostic plots of pHpzc – one of the very crucial parameters in adsorption, which determine the nature of the surface of a dispersed solid phase at a solid-electrolyte solution interface”

Page 7, Line 12. We removed ‘was’

Page 8, Line 44. We revised the phrase after ‘...proportional to...’ into “It was reasonable to ascribe the increasing removal percentage of ibuprofen to enlarging the number of active sites by adding more quantity of MPC in aqueous solution”

Page 9 Line 24. We rewrite as ‘...due to competition between cations and active site seizure of inorganic cations’, as follows: “This may be due to migration of Na⁺ cations to the active sites of adsorbent”.

Page 9, Line 29. We used this word ‘reduces’

Page 10, Line 28. We rewrote the phrase after the comma as follows: “Although H-donors of these functional groups are easily deprotonated in the strongly basic solution, they can be protected in the strongly acidic media (i.e., pH 3), thus easily stabilize “donor-acceptor” complexes in acidic solution”.

Page 10, Line 43. We used ‘playing’

Page 10, Line 46, We used ‘in which’ instead of ‘which’

Page 10, Line 57, Should be ‘was obtained’

Page 11, Line 38. We wrote the whole sentence starting from ‘moreover...’ and splitted it into two separate sentences. “Moreover, with higher regression constant (0.9689) of Bangham's kinetic model, it is suggested that the intra-particle diffusion mechanism may control the adsorption rate at room temperature”.

Page 12, Line 21. We used ‘briefly’

Reviewer: 2

Comments to the Author(s)

It this work, a mesoporous carbon from MIL-53(Fe) is synthesized for the elimination of IBU from water.

Title:

-is it adsorbability or absorbability? May be it need to be modified as the adsorbent is a porous material.

The authors fully appreciate the reviewer's detection to improve the quality of paper. For the right definition, it should be "adsorbability" instead of "absorbability". According to the reviewer's suggestion, we are pleased to provide the misleading definition, and replacing with new one.

Therefore, we adjusted the title as follows: "A hollow mesoporous carbon from metal-organic framework for robust adsorbability of ibuprofen drug in water".

Again, we are very thankful for your detection.

Introduction:

- "Pharmaceutically bioactive compounds (PBCs) are widely consumed over the world because of their crucial role playing in protecting human's health from the attack of bacteria species". Are the PBCs only used to protect from the attack of bacteria species? what about pain? inflammation? or other diseases?

The authors fully appreciate the reviewer's questions to improve the quality of paper. For these concerns, it is true that PBCs include many roles and effects, not only used to protect from the attack of bacteria species. Therefore, we added more information about the role of PBCs according to the reviewer.

In page 3 (lines 4–7), we rewrote the first sentence of the background section under the **turquoise color** as follows: "Pharmaceutically bioactive compounds (PBCs) are widely consumed over the world because of their crucial role playing in protecting human's health from the attack of bacteria

species, as well as exhibiting a wide range of biological activities (e.g. antifungal, anticancer, antitumor, anti-inflammatory, antioxidant, etc.)”

- The authors mentioned that "With their excellent tailorability and versatile functionalities, the MOFs are applicable as promising catalysts, adsorbents, drug delivery systems 20–24." However, in this references there are no articles related with the drug delivery. They need to complete this information.

The authors fully appreciate the reviewer’s detection to improve the quality of paper. For these concerns, we modified a recent review discussing on the application of MOFs in the drug delivery field.

In page 3 (lines 40- 42), there are six articles provided to mention about these concerns:

1. I. A. Lázaro and R. S. Forgan, *Coord. Chem. Rev.*, 2019, 380, 230–259.
2. T. V Tran, H. T. N. Le, H. Q. Ha, X. N. T. Duong, L. H.-T. Nguyen, T. L. H. Doan, H. L. Nguyen and T. Truong, *Catal. Sci. Technol.*, 2017, 7, 3453–3458.
3. H. T. N. Le, T. V Tran, N. T. S. Phan and T. Truong, *Catal. Sci. Technol.*, 2015, 5, 851–859.
4. N. D. Trinh and S.-S. Hong, *J. Nanosci. Nanotechnol.*, 2015, 15, 5450–5454.
5. Ş. S. Bayazit, S. T. Danalıoğlu, M. Abdel Salam and Ö. Kerkez Kuyumcu, *Environ. Sci. Pollut. Res.*, 2017, 24, 25452–25461.
6. R. W. Flaig, T. M. Osborn Popp, A. M. Fracaroli, E. A. Kapustin, M. J. Kalmutzki, R. M. Altamimi, F. Fathieh, J. A. Reimer and O. M. Yaghi, *J. Am. Chem. Soc.*, 2017, 139, 12125–12128.

- What is the concentration of IBU in wastewater? is it similar to the one used in this study? What is the pH of this wastewater? Why is important to eliminate IBU from water better than other pharmaceutical contaminants?

The authors fully appreciate the reviewer’s detection to improve the quality of paper. For these concerns, we are pleased to respond according to our best knowledge as follows. For concentration of IBU in wastewater, Miege et al. reported a database to quantitatively assess the occurrence and removal efficiency of pharmaceuticals and personal care products (PPCPs) in wastewater treatment processes (WWTPs) from many scientific publications, which IBP concentrations in the effluents leaving several sewage-treatment plants were found to be between 0.17 and 59.2 ug/L.[9]. Moreover,

ibuprofen concentration reported in effluents in France and Sweden were 7.11, and 85 $\mu\text{g/L}$, respectively [10]. The IBU concentration varies and depends on the geographically polluted regions, and contaminated environment (stream, river, etc.). For example, the water column of Lake Greifensee (Switzerland) was mean 1.3 $\mu\text{g/L}$ for ibuprofen [11], while this number in the Hölje River, Sweden was from 0.12 to 2.2 $\mu\text{g/L}$ [12].

It is extensively used as non-prescription medicine, with an annual consumption of several hundreds of tons in developed countries. For example, in France, UK, and Spain, it was reported that the volume of pharmaceutically active compounds sold in different countries was great, for ibuprofen, at more than 240, 330, and 276 tons only in 2004 [10]. Moreover, excretion rate of ibuprofen is high (up to 8%) with an incomplete metabolite, likely leading to the penetration of ibuprofen into soil, aquatic media, even human's food source, therefore, it is important to eliminate IBU from water better than other pharmaceutical contaminants [10].

In this study, we conducted the experiments in the range of ibuprofen concentration from 5–20 mg/L , which is so far higher than the fate and occurrence of ibuprofen in geographically polluted regions, and contaminated environment (stream, river, etc.). This concentration range is reported in many works. For example, Sung et al. also reported that the adsorption of ibuprofen at the concentration 5–20 mg/L . However, it is unnecessary to employ at higher such concentration due to low solubility of ibuprofen (21 mg/L) [2]. To obtain the higher concentration, it may be under the use of surfactants such as polyvinyl pyrrolidone (PVP), sodium lauryl sulphate (SLS) or other methods according to recent studies [3–5]. However, we solely carried out the experiments in water without surfactants, therefore, it is reported to be 5–20 mg/L for ibuprofen concentration.

The pH values of ibuprofen-contaminated wastewater can vary in wide range [13]. In this study, however, we investigated the effect of pH on the ibuprofen adsorption and to obtain the highest removal efficiency. Therefore, based on the investigation of adsorption of ibuprofen by MPC, the experiments need to be performed at the optimal pH condition (at pH 3) to reach the maximum adsorption capacity (206.5 mg/g).

With analysis mentioned above, we inserted two passages along with the respective references as follows:

In the introduction, pages 3 (starting from line 25) with a yellow highlight:

“For concentration of IBU in wastewater, Miege et al. reported a database to quantitatively assess the occurrence and removal efficiency of pharmaceuticals and personal care products (PPCPs) in wastewater treatment processes (WWTPs) from many scientific publications, which IBU concentrations in the effluents leaving several sewage-treatment plants were found to be between 0.17 and 59.2 ug/L [9]. Moreover, ibuprofen concentration reported in effluents in France and Sweden were 7.11, and 85 ug/L, respectively [10]. The IBU concentration varies and depends on the geographically polluted regions, and contaminated environment (stream, river, etc.). For example, the water column of Lake Greifensee (Switzerland) was mean 1.3 µg/L for ibuprofen [11], while this number in the Höje River, Sweden was from 0.12 to 2.2 µg/L [12].

It is extensively used as non-prescription medicine, with an annual consumption of several hundreds of tons in developed countries. For example, in France, UK, and Spain, it was reported that the volume of pharmaceutically active compounds sold in different countries was great, for ibuprofen, at more than 240, 330, and 276 tons only in 2004 [10]. Moreover, excretion rate of ibuprofen is high (up to 8%) with an incomplete metabolite, likely leading to the penetration of ibuprofen into soil, aquatic media, even human’s food source, therefore, it is important to eliminate IBU from water better than other pharmaceutical contaminants [10].”

Experimental

-The authors measured the IBU concentration by UV-vis but at which wavelength??

The authors fully appreciate the reviewer’s detection to improve the quality of paper. We added the wavelength at 222 nm for ibuprofen determination by UV-Vis spectroscopy.

In page 5, line 5, a modification text was inserted under the **turquoise color**.

-Is there any work in bibliography where the elimination of IBU in wastewater is studied? What is the % of eliminated IBU? For example: “Ultrasonic treatment of water contaminated with ibuprofen”. *Water Research* 42 (16): 4243-4248, 2008. or "Synthesis of composite iron nano adsorbent and removal of ibuprofen drug residue from water" *Journal of Molecular Liquids* Volume 219, July 2016, Pages 858-864. Among others.

For the previous studies on the elimination of IBU in wastewater, we agree with the reviewer that there is a lack of review about previous studies on the elimination of IBU in wastewater. We therefore attempted to refer to two references that the reviewer provided. We also hope that these documents help the readers find a useful source.

In page 3, lines 30–35, we inserted new bibliography in studies on elimination of IBU in wastewater along with appropriate references under the **turquoise color** as follows:

“For example, Méndez-Arriaga et al. used ultrasonic waves as a means of treatment for the degradation of water contaminated with ibuprofen, obtained the promising results, at 98% within 30 min [1]. Meanwhile, Ali et al. reported the green synthesis of a composite nanoscaled-iron as new generation adsorbent for 92% removal of ibuprofen upon natural water resource conditions (pH 7, low iron dose and agitation time) [6]”

- MIL-53 is used as a source of Fe, as Fe can be used for the "Chemisorption" of IBU. But, how can the authors confirm that is a chemisorption? There is a complete characterization of the MIL-53, and the MPC, but nothing about the IBU loaded MPC. Can the authors add the FTIR, ATG, SEM, BET, DXRP of the loaded matrix?

The authors are very thankful for your comments, and we think that these concerns are very intriguing. Based on our best knowledge, we are pleased to respond as follows:

Chemisorption is characterized by the interaction of chemical groups between adsorbent and adsorbate. It is understandable that the more chemical functional groups (acidic, laconic, phenolic, basic groups) exist on the surface of MPC, the better the adsorption of ibuprofen is facilitated to occur. Therefore, we determined the quality and quantity of functional groups on the surface of MIL-53 (Fe) and MPC via Boehm titration. According to Table 1, it is revealed that the number of functional groups including total oxygenated (2.2 mmol/g) and basic (0.85 mmol/g) groups for MPC. Above functional groups may contribute to enhancing the adsorbability of MPC towards ibuprofen. These results are again supported by the kinetic studies, which the pseudo-second-order model was found to be the most suitable, demonstrated by the extremely high coefficient of determination ($R^2 = 0.9994$) of the linear plot (Table 2). This model assumes that adsorption of ibuprofen over MPC can be chemisorption via electrostatic attraction between adsorbent and adsorbate [14]. Moreover, there are many works proved the adsorption of ibuprofen and other drugs onto MOFs-derived MPC

was the chemisorption process with the crucial role of functional groups [2,15–18]. Therefore, we argue that the chemisorption may be a dominance of ibuprofen adsorption in this study.

With analysis mentioned above, confirming chemisorption was incorporated into the manuscript as follows:

In page 11, starting from line 25, we inserted new passage with a yellow highlight:

“Note that chemisorption is characterized by the interaction of chemical groups between adsorbent and adsorbate. It is understandable that the more chemical functional groups (acidic, laconic, phenolic, basic groups) exist on the surface of MPC, the better the adsorption of ibuprofen is facilitated to occur. In fact, we determined the quality and quantity of functional groups on the surface of MIL-53 (Fe) and MPC via Boehm titration. According to Table 1, it is revealed that the number of functional groups including total oxygenated (2.2 mmol/g) and basic (0.85 mmol/g) groups for MPC. Above functional groups may contribute to enhancing the adsorbability of MPC towards ibuprofen. Tung et al. also demonstrated that surface functional groups are a key factor for adsorption of ibuprofen [19]. Moreover, there are many works proved the adsorption of ibuprofen and other drugs onto MOFs-derived MPC was the chemisorption process with the crucial role of functional groups [2,15–18]. Consequently, we argue that the chemisorption may be a dominance of ibuprofen adsorption in this study.”

Table 1. Surface groups (mmol/g) obtained from Boehm titrations and textual properties of MIL-53 (Fe) and MPC

No	Materials	MIL-53 (Fe)	MPC
1	Carboxylic groups (mmol/g)	0	1.05
2	Lactonic groups (mmol/g)	0	0.5
3	Phenolic groups (mmol/g)	0	0.65

4	Total oxygenated groups (mmol/g)	0	2.2
5	Total basic groups (mmol/g)	0	0.85
6	S_{BET} (m ² /g)	7.6	199.0
7	Magnetization saturation (emu/g)	0	6.3

Table 2. Kinetic constants for the adsorption of ibuprofen on MPC

Kinetic models	Parameters	Unit	Value
Pseudo first-order	k_1	min ⁻¹ /(mg/L) ^{1/n}	0.0154
	q_e	mg/g	70.74
	R^2		0.9946
Pseudo second-order	k_2	g/(mg.min)x10 ⁴	2.97
	q_e	mg/g	99.21
	$H = k_2q_e^2$		2.926
	R^2		0.9994
Elovich	β	g/mg	0.0236
	α	mg/(g.min)	30.38
	R^2		0.9543
Bangham	k_B	mL/(g/L)	0.0429
	α_B		0.5528
	R^2		0.9689

Moreover, to confirm the high stability of MPC materials, the XRD spectra of the pristine and ibuprofen-loaded MPC were used. As can be observed from Fig. 1, the main peak at 44.5° of ibuprofen-loaded MPC still maintained, and existed at the similar position as pristine MPC, proving that the structure of MPC is stable beneath ibuprofen adsorption conditions in aqueous media.

Fig. 1. XRD (a) spectra of pristine and ibuprofen-loaded MPC

- In all the data there are no errors, have the authors checked the reproducibility of these results? They have to add the error values and the error bars in all the figures.

The authors highly appreciate the reviewer's comment to improve the quality of the manuscript. According to the reviewer's suggestion, we inserted new figures (Pages 22, 23) with error bars.

In pages 22, 23, we replaced the old with new figures, as follows:

Fig. 4. Effect of dosage (a), ionic strength (b), solution pH (c), contact time (d), concentration (e), and temperature (f) on the adsorption of ibuprofen onto MPC material

Fig. 5. Recyclability study of MPC material

References

- [1] F. Méndez-Arriaga, R.A. Torres-Palma, C. Pétrier, S. Esplugas, J. Gimenez, C. Pulgarin, Ultrasonic treatment of water contaminated with ibuprofen, *Water Res.* 42 (2008) 4243–4248.
- [2] B.N. Bhadra, I. Ahmed, S. Kim, S.H. Jung, Adsorptive removal of ibuprofen and diclofenac from water using metal-organic framework-derived porous carbon, *Chem. Eng. J.* 314 (2017) 50–58.
- [3] S.G. Potta, S. Minemi, R.K. Nukala, C. Peinado, D.A. Lamprou, A. Urquhart, D. Douroumis, Preparation and characterization of ibuprofen solid lipid nanoparticles with enhanced solubility, *J. Microencapsul.* 28 (2011) 74–81.
- [4] K. Stoyanova, Z. Vinarov, S. Tcholakova, Improving Ibuprofen solubility by surfactant-facilitated self-assembly into mixed micelles, *J. Drug Deliv. Sci. Technol.* 36 (2016) 208–215.
- [5] T.K. Giri, H. Badwaik, A. Alexander, D.K. Tripathi, Solubility enhancement of ibuprofen in

- the presence of hydrophilic polymer and surfactant, *Int. J. Appl. Biol. Pharm. Technol.* 1 (2010) 793–800.
- [6] I. Ali, Z.A. AL-Othman, A. Alwarthan, Synthesis of composite iron nano adsorbent and removal of ibuprofen drug residue from water, *J. Mol. Liq.* 219 (2016) 858–864.
- [7] H. Guedidi, L. Reinert, J.-M. Lévêque, Y. Soneda, N. Bellakhal, L. Duclaux, The effects of the surface oxidation of activated carbon, the solution pH and the temperature on adsorption of ibuprofen, *Carbon N. Y.* 54 (2013) 432–443.
- [8] H.-H. Cho, H. Huang, K. Schwab, Effects of Solution Chemistry on the Adsorption of Ibuprofen and Triclosan onto Carbon Nanotubes, *Langmuir.* 27 (2011) 12960–12967.
- [9] C. Miège, J.M. Choubert, L. Ribeiro, M. Eusebe, M. Coquery, Removal efficiency of pharmaceuticals and personal care products with varying wastewater treatment processes and operating conditions—conception of a database and first results, *Water Sci. Technol.* 57 (2008) 49–56.
- [10] S.C. Monteiro, A.B.A. Boxall, Occurrence and fate of human pharmaceuticals in the environment, in: *Rev. Environ. Contam. Toxicol.*, Springer, 2010: pp. 53–154.
- [11] C. Tixier, H.P. Singer, S. Oellers, S.R. Müller, Occurrence and fate of carbamazepine, clofibric acid, diclofenac, ibuprofen, ketoprofen, and naproxen in surface waters, *Environ. Sci. Technol.* 37 (2003) 1061–1068.
- [12] A. Nikolaou, S. Meric, D. Fatta, Occurrence patterns of pharmaceuticals in water and wastewater environments, *Anal. Bioanal. Chem.* 387 (2007) 1225–1234.
- [13] J.L. Packer, J.J. Werner, D.E. Latch, K. McNeill, W.A. Arnold, Photochemical fate of pharmaceuticals in the environment: Naproxen, diclofenac, clofibric acid, and ibuprofen, *Aquat. Sci.* 65 (2003) 342–351.
- [14] K.L. Tan, B.H. Hameed, Insight into the adsorption kinetics models for the removal of contaminants from aqueous solutions, *J. Taiwan Inst. Chem. Eng.* 74 (2017) 25–48.
- [15] H.J. An, B.N. Bhadra, N.A. Khan, S.H. Jhung, Adsorptive removal of wide range of pharmaceutical and personal care products from water by using metal azolate framework-6-derived porous carbon, *Chem. Eng. J.* 343 (2018) 447–454.

- [16] M. Sarker, J.Y. Song, S.H. Jung, Adsorptive removal of anti-inflammatory drugs from water using graphene oxide/metal-organic framework composites, *Chem. Eng. J.* 335 (2018) 74–81.
- [17] B.N. Bhadra, S.H. Jung, A remarkable adsorbent for removal of contaminants of emerging concern from water: Porous carbon derived from metal azolate framework-6, *J. Hazard. Mater.* 340 (2017) 179–188.
- [18] P.W. Seo, N.A. Khan, S.H. Jung, Removal of nitroimidazole antibiotics from water by adsorption over metal–organic frameworks modified with urea or melamine, *Chem. Eng. J.* 315 (2017) 92–100.
- [19] T.X. Bui, H. Choi, Adsorptive removal of selected pharmaceuticals by mesoporous silica SBA-15, *J. Hazard. Mater.* 168 (2009) 602–608.

Yours Sincerely,

Trinh Duy Nguyen, Corresponding author

NTT Institute of High Technology, Nguyen Tat Thanh University

298-300A, Nguyen Tat Thanh Street, District 4, Ho Chi Minh City, Vietnam

E-mail: ndtrinh@ntt.edu.vn

Phone: +84-971-275-356